# INSTRUCTSCORE: Explainable Text Generation Evaluation with Fine-grained Feedback

**Wenda Xu[¶], Danqing Wang[¶], Liangming Pan[¶], Zhenqiao Song[¶],**
**Markus Freitag[†], William Yang Wang[¶], Lei Li[‡]**

[¶]University of California, Santa Barbara, [†]Google Research, [‡]Carnegie Mellon University

{wendaxu, danqingwang, liangmingpan, zhenqiao, william}@cs.ucsb.edu

freitag@google.com    leili@cs.cmu.edu

## Abstract

Automatically evaluating the quality of language generation is critical. Although recent learned metrics show high correlation with human judgement, these metrics do not provide explicit explanation of their verdict, nor associate the scores with defects in the generated text. To address this limitation, we present INSTRUCTSCORE, **a fine-grained explainable evaluation metric** for text generation. By harnessing both explicit human instruction and the implicit knowledge of GPT-4, we fine-tune a text evaluation metric based on LLaMA, producing both a score for generated text and a human readable diagnostic report. We evaluate INSTRUCTSCORE on a variety of generation tasks, including translation, captioning, data-to-text, and commonsense generation. Experiments show that our 7B model surpasses all other unsupervised metrics, including those based on 175B GPT-3 and GPT-4. Surprisingly, our INSTRUCTSCORE, even without direct supervision from human-rated data, achieves performance levels on par with state-of-the-art metrics like COMET22, which were fine-tuned on human ratings.

## 1 Introduction

Although large language models (LLMs) have led to significant progress in various natural language tasks  (Brown et al., 2020; Ouyang et al., 2022; Touvron et al., 2023), it remains a challenge to automatically evaluate the quality of text generation across versatile tasks. Traditional word overlap metrics, such as $n$-gram matching, BLEU (Papineni et al., 2002), and chrF (Popović, 2015), along with distance-based metrics like TER (Snover et al., 2006) do not best align with human experts' judgements (Freitag et al., 2021a). They primarily focus on surface form differences between reference and candidate texts (Freitag et al., 2020). On the other hand, recent learned metrics such as BERTScore (Zhang et al., 2019), BLEURT (Sellam et al., 2020a), COMET (Rei et al., 2022) and

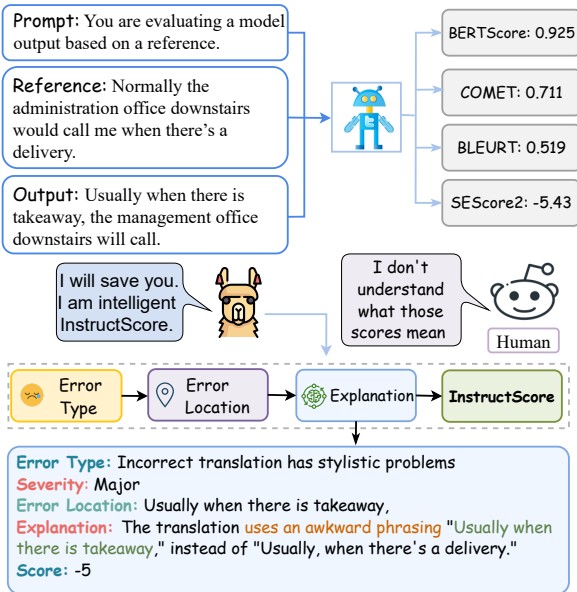

Figure 1: INSTRUCTSCORE generates a comprehensive error diagnostic report for text generation tasks, including error type, location, severity label, and explanation. Based on this report, INSTRUCTSCORE counts the number of major errors (each worth $-5$) and minor errors (each worth $-1$), ultimately assigning a final score.

SEScore (Xu et al., 2022b,a) show a higher correlation with humans on text generation tasks. However, all these metrics produce a single numerical score. These learned metrics lack interpretation of predictions nor link the scores with individual defects in the candidate text.

How can we devise a fine-grained explanation-based text generation metric capable of pinpointing concrete error locations, identifying error types, assigning severity labels, and justifying the final score—all simultaneously without relying on human-annotated data. In this paper, we propose INSTRUCTSCORE, a method to learn an explainable text generation metric without using human annotated ratings. InstructScore provides both a numerical score and a natural language error explanation. To this end, we first extract latent evaluation knowledge from an instruction-following

model, such as GPT-4 (OpenAI, 2023), to construct a synthetic dataset with a predetermined explanation structure. Next, we determine a range of explanation failure modes and devise automated feedback to meta-evaluate error explanations. Finally, we further fine-tune INSTRUCTSCORE model on self-generated outputs that optimize feedback scores, resulting in diagnostic reports that are better aligned with humans.

We have conduct experiments on a variety of text generation tasks: *machine translation*, *table-to-text*, *image captioning*, *commonsense genteration*, and *keyword-to-dialogue generation*. Our experimental findings show that the unsupervised INSTRUCTSCORE outperforms prior strong baselines on all these tasks. It achieves the best results for the unseen *keyword-to-dialogue* generation task. Surprisingly, INSTRUCTSCORE surpasses the supervised BLEURT in 6 out of 9 directions and closely matches state-of-the-art COMET22 in machine translation. Furthermore, we identify a range of failure modes and design an automatic pipeline to pinpoint explanation failures. Our refinement step improves human score by 13.7%, leading to a more accurate alignment with human judgment.

Our INSTRUCTSCORE enjoys the following advantages: (i) **Compact yet competitive**: INSTRUCTSCORE's 7B version displays strong performance compared to metrics based on closed-source 175B LLMs. (ii) **Explainable**: INSTRUCTSCORE provides natural language explanations to justify numerical scores. (iii) **Generalizable**: The unsupervised training pipeline does not require human-annotations, making it easily adaptable to different domains and tasks.

## 2 Related Work

**Learned Evaluation Metrics** Supervised metrics optimize performance by directly fine-tuning human rating data, such as COMET (Rei et al., 2020) and BLEURT (Sellam et al., 2020a), as shown by Rei et al. (2020) and Sellam et al. (2020a). However, human rating data is often unavailable. Unsupervised metrics use different learning objectives or heuristics on embeddings, such as BERT for greedy matching and coverage scoring (Zhang et al., 2019), or sequence-to-sequence models for probability estimation (Thompson and Post, 2020; Yuan et al., 2021). SEScore (Xu et al., 2022b) and SEScore2 (Xu et al., 2022a) train a regression model by synthesizing human-like errors from raw text and using

either a pre-trained natural language inference or a multilingual masked prediction model to attach error severity score. Supervised metrics can arguably attain higher correlations with human judgments (Freitag et al., 2021b, 2022), while unsupervised metrics, such as SEScore (Xu et al., 2022b) and BERTScore (Zhang et al., 2019), exhibit greater levels of generalization. However, none of these approaches offer an explanation for the resulting scores, rendering the decision-making processes obscure and less trustworthy. In this paper, we generate a diagnostic report to provide detailed explanations to support metric's final decisions.

**Explainable Evaluation Metric.** Recent demand for explainability in evaluation metrics has grown significantly. Freitag et al. (2021a) introduce a multi-dimensional human evaluation (MQM) framework for machine translation, while Leiter et al. (2022) investigates key characteristics of explainable metrics. Several metrics derived from those frameworks enhance explainability by differentiating error severity (Lu et al., 2022; Xu et al., 2022b,a; Perrella et al., 2022). Other efforts focus on explanatory aspects of text generation metrics, like error locations (Zerva et al., 2022) and multi-dimensional assessment (Zhong et al., 2022). Despite progress, explanations remain unclear. Researchers also explore LLMs' potential in evaluation, as demonstrated by Fu et al. (2023), but suffers from a lack of explanation. Kocmi and Federmann (2023) and Liu et al. (2023) find large models like GPT-3.5 on system-level can correlate to humans and generate rationales. However, these generated rationales are free-form and may not necessarily align with human judgements (Zheng et al., 2023). In our work, we explicitly refine INSTRUCTSCORE to produce explanations that align with human.

## 3 Problem Definition

Our goal is to learn an explainable metric model that not only predicts the quality score of candidate text comparing to a reference but also generates a diagnostic report in natural language. Specifically, INSTRUCTSCORE assesses the quality of $x$ regarding a reference $r$ by generating an informative diagnostic report, which includes the details about error location $l$, error type $t$, severity level $se$, and explanation $e$ that are associated with the identified error. INSTRUCTSCORE consists of a score predictor and an explanation generator (Exp-Generator) which learns a function $f \colon (\mathbf{x}, \mathbf{y}) \rightarrow \{(l, t, se, e)_i\}_{i=1}^{n}$

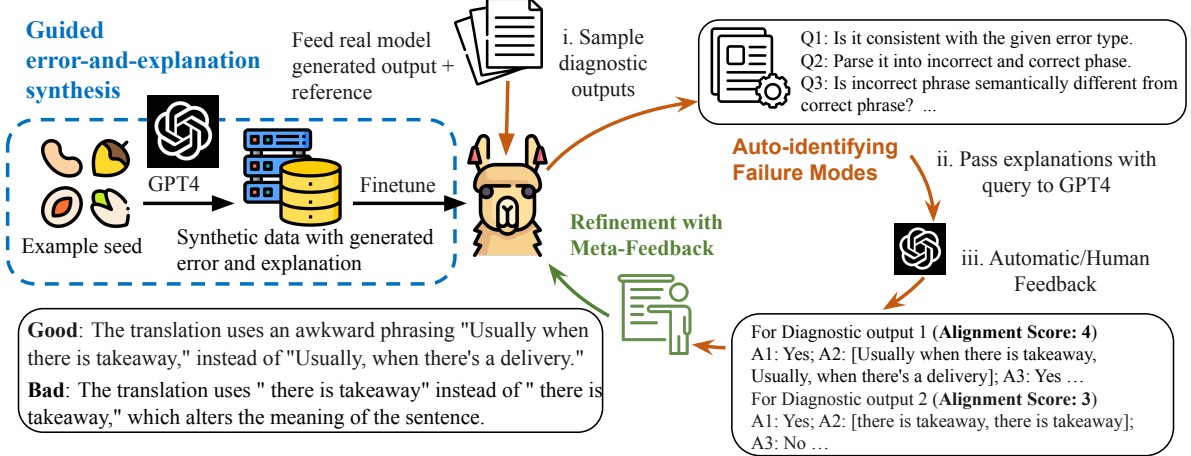

Figure 2: Our INSTRUCTSCORE pipeline consists of three components: First, we construct synthetic data from GPT-4 and use it to fine-tune a 7B LLAMA model. Second, we sample from real-world machine-generated distribution to trigger INSTRUCTSCORE's failure modes. We query GPT-4 on each failure mode and gather automatic feedback. Third, we select explanations that are most aligned with human to further fine-tune LLaMA model. Step 2 and 3 can be repeated to iteratively refine the model output.

with $n$ number of errors. However, such human annotated mapping data for most text generation tasks is scarce due to limited human resources and high annotation costs. To this end, we propose a data construction method to automatically generate high-quality pseudo data to learn $f$.

## 4 The INSTRUCTSCORE Approach

INSTRUCTSCORE assesses the quality of generated texts based on an explainable diagnostic report. Building upon this report, INSTRUCTSCORE provides an intuitive way to comprehend a model's generation capability, resulting in easier comparison among different models. In particular, we begin by extracting concise yet representative explainable knowledge from a large-scale instruction-following model, which is then utilized to train our Exp-Generator. After carefully analyzing the diagnostic reports produced by our Exp-Generator, we summarize common failure modes in diagnostic report and ask GPT-4 to identify them. Then we transform the GPT-4's feedback into alignment scores using our predefined criteria. Finally, we select diagnostic reports that have the highest alignment scores, and further finetune our Exp-Generator on those self-refined outputs. The overall framework is illustrated in Figure 2.

The quality score $s$ for each candidate $y$ is determined based on the number of errors and their severity labels in the diagnostic report. Minor errors are given a score of $-1$ and major errors are given a score of $-5$. These penalties for errors are weighted to calculate the final score. Similar to pre-

vious practices (Freitag et al., 2021a), our metric identifies up to five errors per sentence.

### 4.1 Learning with Guided Error and Explanation Synthesis

We leverage GPT-4 to extract representative explainable knowledge that can greatly contribute to the subsequent Exp-Generator learning process.

Specifically, we collected raw sentences in the target language from diverse domains and topics via GPT-4 (Details are included in Section 5.1 and Appendix A), resulting in data across diverse tasks. This corpus is used as the starting point to inject errors. Then, we prompt GPT-4 to synthesize designated generation errors, as shown in Table 1.

For each text, we specify the number of errors, error types, and severity labels, and ask GPT-4 to generate a candidate output with the specified error descriptions and 2) an explanation for this error annotation. If an evaluation task is multi-dimensional, error types will be separately assigned to each dimension (An example is included in the Appendix). Benefiting from the large-scale pre-training process, GPT-4 is able to generate diverse errors and meet the requirements with specified instructions. To avoid the model's over-reliance on the lexical and structural similarities between the candidate and raw text, we request GPT-4 to rephrase the raw text sentence to construct a pseudo-reference sentence. By specifying error type $t$, severity label $se$, and raw text, GPT-4 is able to generate a synthetic error sentence $x$ with annotated error location $l$ and a pseudo reference $y$ with explanation $e$. Therefore, we can construct synthetic data that reflects

*Prompt for GPT4*:
The correct translation is, "**The art of writing for educational publications involves striking a delicate balance between providing enough detail to be useful and overwhelming the reader with too much information.**" Please give **0 minor errors and 2 major errors**. For each error, give me an error type, a major/minor label, an error location, and an explanation. The format is Paraphrase correct translation:
Incorrect Translation:
Error type 1: **Translation includes information not present in the correct translation**
Major/minor: **Major**
Error location 1:
Explanation for error 1:
Error type 2: **Problems with grammar, other than orthography**
Major/minor: **Major**
Error location 2:
Explanation for error 2:

---

*GPT4 Output*:
Error type 1: Translation includes information not present in the correct translation
Major/minor: Major
Error location 1: "...giving enough examples..."
Explanation for error 1: The correct translation talks about providing enough detail, not specific examples, which changes the meaning of the sentence.
Error type 2: Problems with grammar, other than orthography
Major/minor: Major
Error location 2: "...making the reader bored with too many details."
Explanation for error 2: The correct translation mentions overwhelming the reader with too much information, not making them bored with too many details. This change in phrasing alters the meaning of the sentence.

Table 1: This prompt was used to generate synthetic data to fine-tune the LLAMA model, specifically GPT-4. The model was provided with the number of errors in a sentence. For each error annotation, we pre-define the error type and a severity label. GPT-4 was then required to identify the error locations and provide explanations for each error annotation. The texts **highlighted** are variables for each synthesized generation. The complete example can be found in Appendix Table 29.

the relationship between $(x, y)$ and $(t, l, se, e)$.

We train our Exp-Generator by utilizing the constructed data. Particularly, we use LLaMA as the initialization of the Exp-Generator since it is open-sourced and performs well in both understanding and generation tasks. Then we train our Exp-Generator by taking the pseudo reference $y$ and candidate $x$ as the input, and the diagnostic report including the corresponding error type $t$, error location $l$, severity label $se$ and the explanation $e$ as the output. A concrete example can be found in Figure 2. Accordingly, our training objective can be defined as follows:

$$\mathcal{L}(t, l, se, e, x, y) = -\log P(t, l, se, e | y, x; \theta) \quad (1)$$

where $\theta$ is the trainable parameter of the Exp-Generator.

## 4.2 Auto-Identifying Failure Modes of Metric Output

The diagnostic report plays an important role in text quality explanation. However, the above trained model is not guaranteed to produce sensible explanations – those incorrect explanations are referred to as failure modes. We categorize failure modes into global and local levels. A global failure invalidates all four fields: error type, error location, major/minor and explanation. A local failure only affects a specific field, like error type.

In Table 2, we define six scenarios M1-M6 for local failures and four scenarios G1-G4 for global failures. We demonstrate one failure mode M4 in Table 3. Concrete examples for each failure mode are included in the Appendix Table 13, 14, 15, 16, 17, 18, 19, 20, 21, 22. A special case is when the method outputs an annotation containing no error. In this situation, we need to verify if this is accurate (A concrete example can be found in Appendix Table 35). If errors are present, the diagnostic report is incorrect.

Ideally, a human annotator can provide the most accurate judgment for detecting each failure mode. However, obtaining annotations from humans for every instance of a diagnostic report is infeasible. As an alternative, we leverage GPT-4's capabilities in information extraction, parsing, and semantic understanding (OpenAI, 2023) to convert complex requirement queries into simple Yes/No questions.

Specifically, we prompt GPT-4 to parse the explanation into incorrect and correct phrase pairs and extract the error span from the error location. To address hallucinations from error location (M3) and explanation (M4), we verify if our parsed error span is present in the candidate sentence. If one error annotation contains multiple incorrect-correct phrase pairs, it indicates multiple errors in one error location (G4). To address G1, we first check if the incorrect phrase is indeed an error. Additionally, we verify if the revision suggestion is in the output. For the remaining M and G aspects, we design specific prompt queries for the GPT-4. A detailed example of the prompt query for checking M1-M6 and G1-G4 can be found in Table 4.

After obtaining local and global failure modes from GPT-4's feedback, we can convert those feedback into alignment scores. We assign a binary

| Fields | Explanation Failure Mode | Description |
|---|---|---|
| | | Local Failure Mode |
| Error Type | Inconsistency to explanation | M1: Error type descriptions are not consistent with explanation |
| Error Location | Inconsistency to explanation
Error location hallucination | M2: Error locations are not consistent with the explanation
M3: Error locations are not referred in the output text |
| Major/Minor | Major/Minor disagreement | M5: Major and minor labels do not correspond to the correct severity levels |
| Explanation | Error location hallucination
Explanation failure | M4: Error locations can not refer to the output text
M6: The explanation is wrong. However, error at a specified location does exist |
| | | Global Failure Mode |
| All 4 Fields | False negative error
Repetition
Phrase misalignment
Mention multiple errors | G1: Error described in the explanation is not an error
G2: One error is mentioned more than once among explanations
G3: Incorrect phrase and correct phrase are not correctly aligned
G4: One error span mentions multiple errors |

Table 2: This table shows common failure modes of the explanation output of first step Exp-Generator (Fine-tuned LLaMA on synthetic data **without refinement**). Local errors are field-specific, which only correspond to the error of the local field. Global errors can affect all four fields, such as error type, error location, major/minor, and explanation. The observation of the failures modes at first step Exp-Generator is the main motivation for us to perform refinement with automatic feedback (See Section 4.3)

---

**INSTRUCTSCORE Input**:
**Reference:** Activities include the "Cool Maker Loves Family" carnival of children's future learning, the first children's visual art festival, the "Book Carnival" event and the "Whole Family Loves Reading" Picture Book Reading activity.
**Output:** Activities include the "love family and create cool" children's future learning carnival, the first children's visual arts festival, the "book carnival" activity, the "family love reading" picture book series activity.

**INSTRUCTSCORE Output**:
**Your Translation contains 5 errors:**
**Error type 5:** Incorrect translation includes information not present in the correct translation
**Major/minor:** Major
**Error location 5:** "whole family loves reading"
**Explanation for error 5:** The incorrect translation adds the phrase "whole family loves reading," which is not present in the correct translation and changes the focus from "families" to "whole families."

Table 3: This is the case where error locations in explanation can not refer to the output text (**M4**). "whole family loves reading" is never mentioned in the output text. Instead, it was mentioned in the reference text. The complete example can be found in Appendix Table 16.

---

**Prompt**: Ref: ...... entered the revolutionary base area of south Jiangxi. Out: ...... entered the old revolutionary district of southern Jiangxi.
Error location 1: "old revolutionary district"
Error type 1: Terminology is non-standard
Explanation 1: The correct term should be "new revolutionary base area" ......
Q1: For each error location, extract the incorrect error location.
Q2: Parse the explanation into either one of the four forms: [incorrect phrase, correct phrase]......
Q3: If A2 is "incorrect phrase to correct phrase", is A2 a correct alignment for reference and output?
Q4: According to the sentence context, is it no-error or minor-error or major-error?
Q5: Is the explanation consistent with the given error type?
Q6: Is error location in explanation?
Q7: Do two error locations mention the same location?

Table 4: Prompt for GPT4 feedback: We asked following questions to determine the correctness and consistency of the explanation. This is a simplified prompt of how we obtained GPT-4 feedback for Table 2 (Complete examples can be found in Appenfix Table 34 and 35).

---

score in each field of the diagnostic output. If one local error is observed, the corresponding field will receive score of 0. If one global error is observed, all four fields corresponding to that annotation will receive 0s. Each diagnostic report has a score of $\frac{\#correct fields}{\#total fields}$, yielding a alignment score between 0 and 1. For example, one candidate sentence can be given four error annotations. For each error annotation, there are four fields: error type, location, major/minor, and explanation, 16 fields in total. If one global and one failure modes are observed, the

corresponding alignment score will be 11/16.

## 4.3 Refinement with Meta-Feedback

We then leverage the well-learned Exp-Generator to produce high-quality pseudo training data to iteratively refine the model performance, which can further benefit final quality score calculation. Specifically, we use hypothsis $h_i$ with reference $k_i$ as the model input and employ sampling strategies to generate diverse diagnostic reports. Due to discrepancies between synthetic data and real world hypothesis-reference pairs $\{x, y\}$ and

$\{h, k\}$ (Sellam et al., 2020a; Xu et al., 2022b), we anticipate real world model hypothesis can trigger diverse failure modes from Table 2. For each input pair $(h_i, k_i)$, we use top p samping to sample $n$ possible diagnostic outputs, denotated as $\{o_1, o_2, ..., o_n\}$. Based on the feedback scores from GPT-4, we keep the diagnostic output that optimizes the alignment score, $o_{aligned} = \{t_{aligned}, l_{aligned}, se_{aligned}, e_{aligned}\}$, and further fine-tune our Exp-Generator. This automatic critiques and self-training pipeline can further encourage our Exp-Generator to generate more accurate diagnostic reports. Based on the human evaluation, the final quality score can reduce failure modes and align outputs with humans better.

$$\mathcal{L}(o_{aligned}, x, y) = -\log P(o_{aligned}|y, x; \theta) \quad (2)$$

## 5 Experiments

In the experiment section, we aim to answer the following research questions: 1) *What is the performance across various tasks within the English language?* 2) *What is the performance across different domains within the same task?* 3) *What is the performance across different evaluation dimensions?* 4) *What is the performance at unseen tasks?* 5) *Given that LLaMA is predominantly trained in English texts, can it effectively evaluate generations in other languages?* 6) *Can we align the diagnostic report with human expectations without requiring extensive human efforts?*

To answer Q1, we tested INSTRUCTSCORE at various tasks, including WMT22 (*Machine Translation*) (Freitag et al., 2022), WebNLG (*Table-to-text*) (Castro Ferreira et al., 2020), Flicker3k (*Captioning*) (Hodosh et al., 2013), BAGEL (*Keyword-to-text*) (Mairesse et al., 2010) and Commongen (*Commonsense text generation*) (Lin et al., 2020). To address Q2, we exampled our method at four diverse domains: *News*, *Conversation*, *Social*, and *E-Commerce* at WMT22. For Q3, we evaluated our method at five evaluation dimensions at WebNLG. For Q4, we evaluated INSTRUCTSCORE at BAGEL benchmark, a task and evaluation dimensions that are unseen in the synthetic data. For Q5, we evaluated our approach to English-to-German translations in order to investigate its multilingual evaluation capabilities. Lastly, we demonstrate our metric with automatic critique and refinement can achieve higher human ratings regarding failure modes.

### 5.1 Experiment Setup

**Baseline and Benchmark** We tested our INSTRUCTSCORE at 1) **WMT22** shared metric task, in two target languages: English and German. WMT22 uses an MQM-based human evaluation procedure (Freitag et al., 2021a). WMT22 has 14 English, 16 German participating Machine Translation systems, with 26,250 and 21,040 human-annotated outputs respectively; 2) **WebNLG20**: The input of the task contains Wikipedia triples, and the output is a natural language text. This benchmark contains 16 participating WebNLG systems with 2832 human-annotated outputs; 3) **Flicker3K-CF**: This benchmark contains 145K binary quality judgment gathered from CrowdFlower over 48K (image, caption) pairs with 1K unique images; 4) **Commongen**: This benchmark contains 2796 model outputs from 6 participating systems. All human annotations are done in pairwise rankings for the same source input. Therefore, we compute ranking accuracy to estimate the metric's performance; 5) **BAGEL**: The input of this task contains keywords and the output is a fluent human conversation. This benchmark contains 202 model outputs. We included top-performing baseline metrics which joined each challenge, including n-gram based metrics: BLEU (Papineni et al., 2002), chrF (Popović, 2015), METEOR (Banerjee and Lavie, 2005) and CIDEr (Vedantam et al., 2015); Unsupervised learned metrics: PRISM (Thompson and Post, 2020), BARTScore (Yuan et al., 2021), BERTScore (Zhang et al., 2019), and SEScore2 (Xu et al., 2022a); LLM-based metrics: GPT3-Dav3 and GPT4(Kocmi and Federmann, 2023); Supervised learned metrics: BLEURT-20 (Sellam et al., 2020a), MaTESe (Perrella et al., 2022), UniTE (Wan et al., 2022) and MetricX XXL.

**Implementation** We utilize GPT-4 as our implicit evaluation knowledge base and LLaMA-7B as our trained initialization[1]. To ensure coverage of diverse text domains, we separately collect a dataset of 10k raw sentences from 100 different domains using GPT-4 for both English and German. Details on the construction of this dataset are provided in Appendix Sec. C. To adapt to specific task domains, we separately gathered 10k sentences for the training datasets of WebNLG17 (Gardent et al.,

---

[1] We include InstructScore based on LLaMA2-7B (Touvron et al., 2023) results in the Appendix Table 12. We demonstrate that varying pertaining initialization can lead to different results at diverse NLG tasks.

| | | Seen Tasks | | | Unseen Task | |
| | | WMT22(Zh→En) | WebNLG20 | Flicker3k | Commongen | BAGEL | Rank |
| | | $\tau/\rho$ | $\tau/\rho$ | $\tau/\rho$ | Acc | $\tau/\rho$ | Avg |
|---|---|---|---|---|---|---|---|
| Supervised | MATESE | 38.9 / 52.8 | - | - | - | - | - |
| | Metric XXL | 42.7 / 58.1 | - | - | - | - | - |
| | COMET-22 | **42.8 / 58.5** | - | - | - | - | - |
| | BLEURT-20 | 36.1 / 43.0 | **40.2 / 63.5** | 24.3 / **35.1** | 39.5 | 22.9 / 32.3 | 2.8 |
| Without Supervision | BLEU | 14.5 / 17.5 | 20.1 / 20.7 | 13.8 / 21.6 | 26.8 | 10.9 / 16.8 | 10.2 |
| | ChrF | 15.4 / 14.7 | 26.6 / 40.0 | 13.3 / 24.5 | 33.0 | 10.8 / 16.8 | 9.0 |
| | METEOR | 16.5 / 18.8 | 25.6 / 36.3 | 13.4 / 23.1 | 33.9 | 12.6 / 14.5 | 9.4 |
| | CIDEr | 18.0 / 22.1 | 26.1 / 31.8 | 15.2 / 29.8 | 32.6 | 15.7 / 23.1 | 7.8 |
| | BERTScore | 31.6 / 37.6 | 32.8 / 50.4 | 17.4 / 24.6 | 38.8 | 17.1 / 28.2 | 5.6 |
| | BARTScore | 22.2 / 24.9 | 33.1 / 56.8 | 17.9 / 22.2 | 37.0 | 20.3 / 20.7 | 6.2 |
| | PRISM | 25.0 / 27.9 | 37.6 / **59.4** | 16.1 / 23.8 | 38.0 | 21.7 / 30.7 | 5.1 |
| | GEMBA-GPT4 | 38.2 / 37.4 | - | - | - | - | - |
| | SEScore2 | 33.0 / 46.4 | 36.8 / 48.4 | 16.7 / 22.2 | 34.4 | 23.3 / 32.5 | 4.9 |
| | INSTRUCTSCORE | **40.3 / 51.9** | **39.5** / 59.0 | **30.1 / 34.6** | **58.2** | **25.6 / 34.2** | **2.0** |

Table 5: We applied segment-level Kendall and Pearson correlation on WMT22, WebNLG20, Flicker3k, and BAGEL, and ranking accuracy for Commongen. INSTRUCTSCORE significantly outperforms all unsupervised metrics in 8/9 directions using William's pairwise significance test (p < 0.05). The top supervised and unsupervised metrics are bolded. Pearson and Kendall correlations are ranked per task, with overall performance being the average rank across tasks. Appendix Table 9 contains ranking details for each correlation and task.

2017), CoCo Captioning (Chen et al., 2015), and CommonGen (Liu et al., 2022). We apply the respective prompts defined in Appendix Tables 29, 30, 31, and 32 to generate synthetic data.

We define four evaluation scenarios: 1) evaluation with reference only; 2) evaluation with reference and additional data; 3) evaluation with reference where the source has different modalities; 4) evaluation with reference and world knowledge. For each scenario, we obtain 10k candidate-reference pairs as input and structured diagnostic reports as output. We train a separate checkpoint for each evaluation scenario, resulting in four checkpoints in total. All models are fine-tuned with language modeling loss with 10k synthetic data. Each model is trained for three epochs, with a learning rate, batch size, and weight decay of 2e-5, 128, and 0, respectively. During the evaluation of each model, we set the temperature to 0 for greedy decoding. Details of our automatic critique and self-training are included in the Appendix Sec E.

**Meta-evaluation** We assess the performance of INSTRUCTSCORE using Segment-level Kendall and Pearson correlations between human and metric output. Kendall Tau-b might favor tie pairs, possibly giving an unfair advantage to certain systems (Deutsch et al., 2023). Pearson, on the other hand, measures linear association. By reporting both complementary results, we can comprehensively understand the metric's performance. We employed three human annotators to assess the

alignment of our model before and after refinement. In particular, the human raters [2] will estimate a binary score based on M1 to M6 and G1 to G4 criteria for each field in the diagnostic report. The total score of a diagnostic report is calculated as $\frac{\#correct fields}{\#total fields}$. The final score is averaged among raters. Details of human evaluation procedures are included in the Appendix Sec F and Table 8.

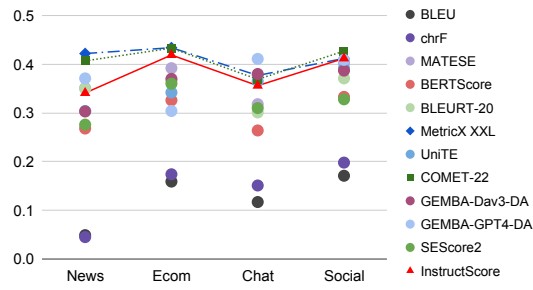

Figure 3: Segment-level Kendall ($\tau$) correlation on Zh-En for different domains of WMT22. We connect points to highlight INSTRUCTSCORE with two other top performing metrics in the figure.

## 5.2 Main Results

**Robust Performance across Tasks** We assess the primary performance of INSTRUCTSCORE for five diverse NLG tasks. As shown in Table 5, INSTRUCTSCORE significantly outperforms all other unsupervised metrics in 8 out of 9 directions, achieving the best overall ranking. All improve-

---
[2]Each human rater is paid at $16.0 per hour, which is above the minimum wage of the local region.

ments are statistically significant by William's pairwise significant test (Graham and Baldwin, 2014) with p < 0.05. Surprisingly, INSTRUCTSCORE even outperforms prior supervised learned metrics that trained over direct assessment data (DA), leading BLEURT20 in 6 out of 9 directions. Compared to GPT4 baseline, INSTRUCTSCORE outperforms GEMBA-GPT4 with 0.021 in Kendall and 0.145 in Pearson correlation. The larger gap in Pearson correlation can be explained by a large set of ties that GEMBA-GPT4 is producing. This will lead to false positive in Kendall correlation. Lastly, we demonstrate that INSTRUCTSCORE can achieve close performance to the supervised learned metrics, MATESE, COMET22 and Metric XXL, that have trained over comprehensive human rating data (DA and MQM), with average 0.012 gap in Kendall correlation and 0.045 in Pearson correlation.

**Robust Performance across Domains** In Figure 3, we further evaluate the performance of INSTRUCTSCORE across different domains. From Kendall correlation analysis, INSTRUCTSCORE surpasses all unsupervised metrics in four domains except GPT-3.5 and GPT-4 baselines at Chat. It achieves comparable performance in *Ecommerce*, *Chat*, and *Social* domains when compared to the leading supervised metrics COMET22 and Metric-XXL. However, its performance is noticeably worse in the *News* domain compared to SOTA COMET22 and Metric-XXL. This gap can be primarily attributed to their supervised data distribution at News domain DA and MQM data.

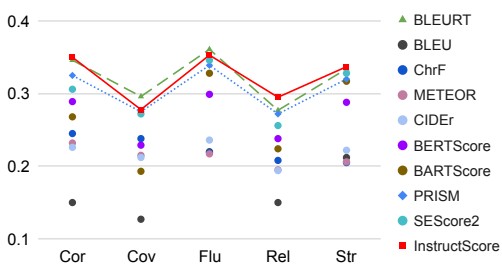

Figure 4: Segment-level Kendall Correlation on WebNLG Data-to-Text generation. Cor, Cov, Flu, Rel, and Str represent Correctness, Coverage, Fluency, Relevance, and Text Structure respectively. We connect points to highlight INSTRUCTSCORE with two other top performing metrics in the figure.

**Robust Performance across Dimensions** Unlike most of metrics which only output a single score, INSTRUCTSCORE can output a score in each evaluation dimension. In Figure 4, we demonstrate

that INSTRUCTSCORE outperforms all unsupervised learned metrics across five different dimensions. Compared to BLEURT, which trained over WebNLG human rating data, INSTRUCTSCORE outperforms best performed BLEURT in three out of five evaluation dimensions. This signifies that INSTRUCTSCORE can be extended to assign a single quality score but extending into multidimensional NLG evaluation.

**Generalization over Unseen Task** Since each NLG task has distinct evaluation criteria, one natural question to ask: Is INSTRUCTSCORE generalizable to the task with unseen data format and evaluation criteria? To verify this question, we use BAGEL benchmark as unseen task since it contains distinct data formats from our training data and contains distinct criteria. From Table 5 and Figure 5, we demonstrate that despite never seen the evaluation criteria of keywords to text, INSTRUCTSCORE achieves the higher Kendall and Pearson correlation compared to BLEURT as well as two of three unseen evaluation dimensions.

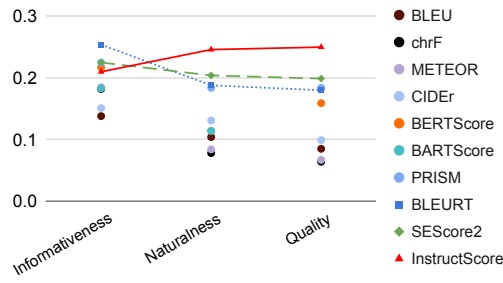

Figure 5: Segment-level Kendall Correlation on different dimensions of BAGEL dialogue generation. We connect points to highlight INSTRUCTSCORE with two other top performing metrics in the figure.

### 5.3 Quantitative Analysis

**Performance at Non-English Language.** In Figure 6, INSTRUCTSCORE outperforms most unsupervised metrics, but not 175B GPT3.5 models at WMT22 English-to-German or supervised counterparts like COMET22 and BLEURT20. We hypothesize that multiple factors contribute to the observed LLaMA performance on non-English texts: (1) limited pretraining data, resulting in weaker pretrained knowledge for other languages, and (2) the task setup requiring alignment between languages due to mixed code text generation (see Appendix Sec D and Table 24). Future research could explore multilingual alignment warmup methods before training on evaluation data.

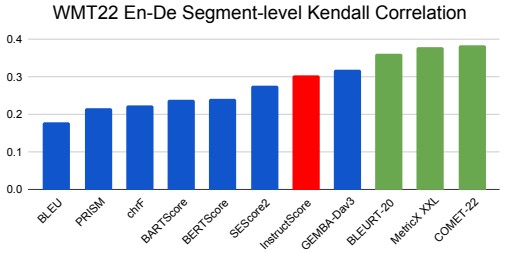

Figure 6: Segment-level Kendall correlation of INSTRUCTSCORE at WMT22 English-to-German. Purple indicates unsupervised learned metrics, while Blue indicates supervised learned metrics.

**Automatic critique and Self-training can improve human Alignment** We conduct human evaluation to assess our metric's alignment before and after self-training. From Figure 7, we demonstrates that INSTRUCTSCORE after automatic-critique and refinement can significantly reduce most of global and local failure modes. In particular, all global failures have more than 50% decreases in occurrences. This signifies that INSTRUCTSCORE has improved over its phrase alignment, error identification and error formats. Moreover, consistency between four fields have all improved, demonstrated by improvements over all M occurrences. We observed that M6 has slight increase. This is due to some of the conversions from global failures into the local failures. In Table 6, we demonstrated that INSTRUCTSCORE can achieve 0.106 absolute human score gains with on par performance at Kendall and Pearson correlation. The detailed human evaluation procedure, individual rater scores and a case study are included in the Appendix Sec F, 10 and 23.

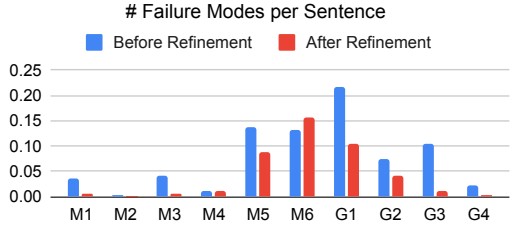

Figure 7: Failure mode occurrence per sentence before and after refinement. Results are averaged by three human raters.

| INSTRUCTSCORE | Kendall | Pearson | Human |
|---|---|---|---|
| Fine-tune | **0.404** | 0.515 | 0.773 |
| Fine-tune+Refinement | 0.403 | **0.519** | **0.879** |

Table 6: We report Segment-level Kendall, Pearson correlation and human score before and after refinement.

**Automatic critique and Self-training can improve precision and recall of INSTRUCTSCORE** We conduct a human evaluation of the quality of INSTRUCTSCORE's annotations (with three annotators). We calculated the precision and recall of the annotations before and after refinement. *Precision = # of correctly annotated error fields / # of INSTRUCTSCORE's labeled error fields. Recall = # of correctly annotated error fields / (# of correctly annotated error fields+the number of error fields that INSTRUCTSCORE missed).* Error fields consist of error type, severity label, error location, and explanations. In Table 7, INSTRUCTSCORE can achieve 77.8% precision and 82.4% recall before refinement. After refinement, INSTRUCTSCORE can improve precision and recall by 11.6% and 3.2%, respectively. In addition, we study the field of explanation in particular, the refinement step can improve the precision of explanation from 75.6% to 86.1% and improve the recall of explanation from 81.9% to 85.0%. **Overall, our finding suggests that 89.4% of InstructScore's output after refinement is correct and it can identify 85.6% of errors produced by the translation system**.

| Human Evaluation | $P_{All}$ | $R_{All}$ | $P_{Exp}$ | $R_{Exp}$ |
|---|---|---|---|---|
| Before refinement | 0.778 | 0.824 | 0.756 | 0.819 |
| After refinement | **0.894** | **0.856** | **0.861** | **0.850** |

Table 7: We report precision and recall before and after refinement on all annotation fields (error type, location, severity label and explanation) by INSTRUCTSCORE, annotated by $P_{All}$ and $R_{All}$. In addition, we include precision and recall on the explanation field only, annotated by $P_{Exp}$ and $R_{Exp}$.

## 6 Conclusion

In this paper, we present a novel framework for explainable text generation evaluation, addressing the existing black-box limitations associated with learned metrics. We define a set of failure modes to regularize the explanations. We empirically demonstrate that INSTRUCTSCORE can be generalized to different domains, tasks and evaluation dimensions, achieving the best ranking compared to other general text generation metrics. Lastly, our refinement from automatic feedback can further improve human alignment score, precision, and recall, by 13.7%, 11.6%, and 3.2%, respectively, leading to a more accurate alignment with human requirements. We released the INSTRUCTSCORE model for public use and open-source the data and codes.

## Limitations

While we have not yet been able to test IN-STRUCTSCORE in a multilingual setting due to the limited availability of human annotation and significant label costs, it has shown promising results in leveraging smaller synthetic and feedback data to fine-tune and refine its performance. In fact, INSTRUCTSCORE has demonstrated superior performance compared to unsupervised baselines, such as BERTScore, BARTScore, and PRISM in high-resource non-English language, such as German. Going forward, we aim to assess IN-STRUCTSCORE's multilingual evaluation capabilities across high, medium, and low-resource languages. As our instructions are in English and the evaluation target is in other language, we plan to enhance INSTRUCTSCORE's mixed code generation and multilingual word alignment abilities by exploring more pretraining and warm-up techniques.

Although our current computing resources restrict our ability to confirm the impacts of model size on performance, future research should investigate model size utilizing scaling law (Kaplan et al., 2020) to uncover potential improvements in failure modes related to larger model sizes.

In the present framework, we introduce a straightforward but efficient refinement process to enhance the alignment of our metric with human judgements. Future research can investigate more advanced techniques, such as incorporating human feedback through reinforcement (Ouyang et al., 2022), for more effective integration of feedback into the training pipeline. More sophisticated approach holds promising potential to further boost the performance of this pipeline.

## Ethics Statement

INSTRUCTSCORE, as an open-source and explainable evaluation metric for text generation, emphasizes transparency and accountability in the evaluation of natural language processing systems. By generating interpretable evaluations and diagnostic reports, it fosters trust among developers and end-users. Moreover, its introduction could propel further innovation in the field of explainable evaluation metrics and make high-quality evaluation tools more accessible. However, it is crucial to ascertain that the interpretations provided by InstructScore do not harbor biases present in the training data, and data privacy and security measures are observed.

The quality improvements that may stem from using InstructScore could be instrumental in diverse applications such as translation services, chatbots, and content creation. Nonetheless, it is vital to monitor these advancements to ensure that they do not inadvertently suppress linguistic diversity. Additionally, the biases that may have been passed on to InstructScore from pre-existing models like GPT4 should be critically examined, and efforts must be made to alleviate biases that could impact language, dialect, or cultural representation.

Finally, the impact of InstructScore on educational and professional writing practices should not be overlooked. As writers and educators might adapt their styles based on algorithmic evaluations, it is essential to balance the quest for higher scores with the preservation of human creativity and the diversity of expression. InstructScore has the potential to be a powerful tool in the evaluation of text generation, but it is imperative that ethical considerations surrounding transparency, accessibility, bias, and societal impact are vigilantly monitored and addressed.

We hired three human raters to annotate IN-STRUCTSCORE's diagnostic reports. We randomly sampled 100 candidate-reference pairs from WMT22 Chinese-to-English direction. All evaluated data does not contain sensitive or explicit languages. There is no risk of exposing raters' identities and they have full knowledge of data usage. All annotators are well-trained with evaluation protocols and are proficient in English. The salary rate is above minimum wage at the local region. All human rating data will be released upon paper acceptance. The detailed human evaluation process is included in the Appendix Sec F.

## 7 Acknowledgement

This work was supported by the National Science Foundation award #2048122. The views expressed are those of the author and do not reflect the official policy or position of the US government.

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

# A  Prompting for Data Generation

In this section, we will discuss the data generation processes using prompts for GPT-4 API. We start from the seed domains, such as News, Technical, Legal and Medical, etc. We query GPT-4 to augment seed domains to 100 domains (See a prompt example in Table 25). For each domain, we query GPT-4 to generate 100 topics (See a prompt example in Table 26). Therefore, we have obtained 10,000 different topics for this process. For each topic, we generate 5 distinct sentences, each with a different length and structure (See a prompt example in Table 27). For each topic, we randomly select one sentence out of five candidates, yielding 10,000 raw text sentences from distinct topics. We start from each raw text to synthesize sentence errors.

Based on the guidelines provided by (Freitag et al., 2021a), we arbitrarily decide the range of errors from 1 to 5. For each raw text, we randomly select a number of errors from 1 to 5. Each synthesized error will have error types that are predefined from MQM guidelines (Freitag et al., 2021a). For each synthesized error, we randomly select one error type that is defined in Table 28 and randomly select major or minor severity labels. Therefore, based on the given raw text, error type, and severity label, GPT-4 needs to generate the location of the error and explanation for the error annotation (See a prompt example in Table 29). To disentangle the model's reliance on sentence structure and lexical overlap, we paraphrase the raw text to yield the pseudo reference. The generated synthetic erroneous sentence will become pseudo model generated text. This completes our synthetic data construction. During the fine-tuning process on the LLAMA model, we input the model with pseudo reference and pseudo candidate text, LLAMA is optimized to generate error type, error location, severity label, and explanation.

# B  Prompting for LLM Feedback

To obtain GPT-4 feedback for INSTRUCTSCORE's generated diagnostic report, we developed two different prompts. If our diagnostic report contains error annotations, we asked 7 questions listed in Table 34 to determine the correctness and consistency of the explanation output. This is how we collected responses for failure modes that we defined in Table 2. if our diagnostic report does not contain error annotation, we directly query GPT-4

**Human Evaluation Instructions**:
M1: Error type descriptions are not consistent with explanation
M2: Error locations are not consistent with the explanation
M3: Error locations are not referred in the output text
M4: Error locations can not refer to the output text
M5: Major and minor labels do not correspond to the correct severity levels
M6: The explanation is wrong. However, error at a specified location does exist.
G1: Error described in the explanation is not an error
G2: One error is mentioned more than once among explanations
G3: Incorrect phrase and correct phrase are not correctly aligned
G4: One error span mentions multiple errors

**The rule is following: local failure mode will only impact local field in the error annotation, like error type, error location or explanation. However, global failure mode is between different error annotations.**

For example, sentence 1 contains 2 errors. You will have 8 fields in total
Error type 1:
Major/minor:
Error location 1:
Explanation for error 1:
Error location 2:
Major/minor:
Error location 2:
Explanation for error 2:

If one M1 and one M2 occur, you will receive score 6/8. If G1 occurs, the entire error annotation 1 or 2 (like error type2, major/minor, location and explanation all become invalid). You will receive score 4/8. If a global failure has an overlap with local failure, you will choose global failure as annotation.

Your annotations will begin from here:

Table 8: This is the human evaluation instruction for human raters.

to validate this claim. If GPT-4 reconfirms with INSTRUCTSCORE 's claim, the feedback score is 1. Otherwise, 0.

## C   Raw Sentence Generation from GPT-4

For the data generation pipeline, we start by prompting GPT-4 with 12 seed domains, such as medical, technology News, etc. We used GPT-4 to augment them to 100 distinct domains. For each domain, we further used GPT-4 to generate 100 topics. In the end, we generate 10k sentences with distinct topics, lengths, and sentence structures (See Appendix Table 25).

## D   Implementation of INSTRUCTSCORE at Non-English Language

We trained a German reference only metric to investigate INSTRUCTSCORE's multilingual capability. In this case, our metric will output explanations in English but quote error locations in German. See an example in Appendix Table 33.

## E   Implementation on Refinement pipeline

We used 18 participating system outputs at WMT20 (Sellam et al., 2020b) to approximate the distribution of the model-generated output. We use 2,000

Chinese-to-English parallel sentences. We randomly select one of the 20 MT systems to translate each source sentence and obtain 2,000 translation outputs in the end. We pair each translation with reference as input to INSTRUCTSCORE and use top p sampling with temperature 0.8 and $p = 0.9$ to generate 8 candidate outputs for each input. We obtained GPT-4's feedback on those 16,000 candidate outputs and formed 35,932 ranking pairs [3]. We selected 4,777 diagnostic outputs which achieved the highest alignments scores to further fine-tune INSTRUCTSCORE.

## F   Human Evaluation Procedure

We conduct a human evaluation on WMT22 Zh-En testing set. We randomly select 100 system outputs from 14 participating systems. Following the prior practice (Freitag et al., 2021a), we hired three graduate students who are proficient in English language annotations. Each rater is trained with our annotation procedure for two hours before the annotations. The detailed human evaluation instruction is included in Table 8. Each rater will give a binary score for each field of error type, error location, major/minor label and explanation. If a global failure

---

[3]We removed tie ranking pairs

| | | MT(Zh→En) Rank($\tau/\rho$) | WebNLG Rank($\tau/\rho$) | Seen Tasks Flicker3k Rank($\tau/\rho$) | Commgen Rank(Acc) | BAGEL Rank($\tau/\rho$) | Unseen Task Rank Avg |
|---|---|---|---|---|---|---|---|
| Supervised | MATESE | 4 / 3 | - | - | - | - | - |
| | Metric XXL | 2 / 2 | - | - | - | - | - |
| | COMET-22 | 1 / 1 | - | - | - | - | - |
| | BLEURT-20 | 6 / 6 | 1 / 1 | 2 / 1 | 2 | 3 / 3 | 2.8 |
| Without Supervision | BLEU | 14 / 13 | 10 / 10 | 8 / 10 | 10 | 9 / 8 | 10.2 |
| | ChrF | 13 / 14 | 7 / 7 | 10 / 4 | 8 | 10 / 8 | 9.0 |
| | METEOR | 12 / 12 | 9 / 9 | 9 / 9 | 7 | 8 / 10 | 9.4 |
| | CIDEr | 11 / 11 | 8 / 8 | 7 / 3 | 9 | 7 / 6 | 7.8 |
| | BERTScore | 8 / 7 | 6 / 5 | 4 / 6 | 3 | 6 / 5 | 5.6 |
| | BARTScore | 10 / 10 | 5 / 4 | 3 / 7 | 5 | 5 / 7 | 6.2 |
| | PRISM | 9 / 9 | 3 / 2 | 6 / 5 | 4 | 4 / 4 | 5.1 |
| | GEMBA-GPT4 | 5 / 8 | - | - | - | - | - |
| | SEScore2 | 7 / 5 | 4 / 6 | 5 / 7 | 6 | 2 / 2 | 4.9 |
| | INSTRUCTSCORE | 3 / 4 | 2 / 3 | 1 / 2 | 1 | 1 / 1 | **2.0** |

Table 9: We rank metrics based on Meta evaluation such as Kendall correlation ($\tau$), Pearson correlation ($\rho$) and Ranking accuracy (Acc) for each task. The final ranking is the average rankings of all Meta-evaluations across five tasks.

| INSTRUCTSCORE | Rater1 | Rater2 | Rater3 |
|---|---|---|---|
| Fine-tune | 0.818 | 0.698 | 0.804 |
| Fine-tune+Refinement | **0.849** | **0.887** | **0.902** |

Table 10: The human scores from three different raters before and after refinement.

has an overlap with local failure, rater will choose global failure as annotation. For each annotated example, rater will count the number of correct fields and divide the total number of fields to obtain a alignment score. The final score of 100 examples is the average of 100 alignment scores. We obtained Kappa index among three raters, 0.388, for inter-rater agreement. In Table 10, it is evident that all raters agree on the improvement in alignment after the refinement process.

| Fields | Explanation Failure Mode | Percent% |
|---|---|---|
| | Local Failure Mode | |
| Error Type | M1: Consistency to explanation | 5.2% |
| Error Location | M2: Consistency to explanation | 1.2% |
| | M3: Error location hallucination | 8.2% |
| Explanation | M4: Error location hallucination | 4.5% |
| | M5: Major/Minor disagreement | 27.6% |
| | Global Failure Mode | |
| Error Type, | G1: No-error | 2.3% |
| Location and | G2: Repetition | 1.8% |
| Explanation | G3: Phrase misalignment | 1.4% |
| | G4: Mention multiple errors | 0.2% |
| | G5: Phrase inconsistency | 2.2% |

Table 11: This table shows explanation failure modes and their corresponding failure occurrence ratios.

## G Qualitative Analysis

In this section, we will display a case study of our generated explanations from INSTRUCTSCORE.

### G.1 Case study of INSTRUCTSCORE's Output

**Failure Modes of INSTRUCTSCORE's Output**
Please check ten failure modes (M1-M6, G1-G4) for INSTRUCTSCORE without refinement (See Table 13, 14, 15, 16, 17, 18, 19, 20, 21, 22).

| | | Seen Tasks | | | Unseen Task |
| | WMT22(Zh→En) | WebNLG20 | Flicker3k | Commongen | BAGEL |
| | $\tau/\rho$ | $\tau/\rho$ | $\tau/\rho$ | Acc | $\tau/\rho$ |
|---|---|---|---|---|---|
| INSTRUCTSCORE (LLaMA) | **40.3 / 51.9** | **39.5 / 59.0** | **30.1 / 34.6** | 58.2 | 25.6 / 34.2 |
| INSTRUCTSCORE (LLaMA2) | 39.6 / 48.9 | 39.0 / 54.1 | 21.1 / 21.8 | **69.9** | **31.0 / 46.2** |

Table 12: We applied segment-level Kendall and Pearson correlation on WMT22, WebNLG20, Flicker3k, and BAGEL, and ranking accuracy for Commongen. INSTRUCTSCORE with LLaMA initialization outperforms LLaMA2 initialization at Translation, data-to-text and captioning. However, INSTRUCTSCORE based on LLaMA2 initialization significantly outperforms LLaMA initialization at Commonsense generation and unseen task text-to-dialogue (BAGEL).

---

**INSTRUCTSCORE Input**:
**Reference:** Recommendation: Swiss legacy; wear time and art on your wrist; OMEGA is elegant and simple, low-key but attractive, suitable for work and formal situations.
**Output:** Recommended reason: Inheriting Switzerland, the art of time is in the wrist, Omega is dignified and elegant, simple and simple, not ostentatious but charming, and can be worn on commuting and formal occasions.

---

**INSTRUCTSCORE Output:**
**Your Translation contains 5 errors:**
**Error type 1:** Incorrect translation does not accurately represent the correct translation
**Major/minor:** Major
**Error location 1:** "Inheriting Switzerland"
**Explanation for error 1:** The phrase "Inheriting Switzerland" does not accurately convey the meaning of "Swiss heritage."
**Error type 2:** Incorrect translation includes information not present in the correct translation
**Major/minor:** Major
**Error location 2:** "the art of time is in the wrist"
**Explanation for error 2:** The phrase "the art of time is in the wrist" introduces information that is not present in the correct translation, which is about the art of time being on the wrist.
**Error type 3:** Incorrect translation is missing content from the correct translation
**Major/minor:** Major
**Error location 3:** "simple and simple"
**Explanation for error 3:** The phrase "simple and simple" is missing the word "elegant" from the correct translation, which should be "simple and elegant."
**Error type 4:** Incorrect translation has stylistic problems
**Major/minor:** Major
**Error location 4:** "not ostentatious but charming"
**Explanation for error 4:** The phrase "not ostentatious but charming" has stylistic problems, as it should be "understated but charming" to match the correct translation.
**Error type 5:** Wrong grammatical register (eg, inappropriately informal pronouns). Internal inconsistency (not related to terminology)
**Major/minor:** Major
**Error location 5:** "commuting"
**Explanation for error 5:** The word "commuting" is not consistent with the correct translation's use of "work" and changes the meaning of the sentence.

---

Table 13: This is the case where error type descriptions are not consistent with explanation (**M1**). Error type 5 should be a word choice error. However, model predicts that this is related to grammatical register.

**INSTRUCTSCORE Input:**
**Reference:** The advantage of building inspection shafts using concrete modules is that the four sides of the modules are of concave groove structures, which are built to form chain locks.
**Output:** The advantage of using concrete modules to build inspection wells is that the upper, lower, left and right sides of the modules are all concave-convex groove structure, and the combined masonry forms a chain lock.

**INSTRUCTSCORE Output:**
**Your Translation contains 4 errors:**
**Error type 1:** Incorrect translation does not accurately represent the correct translation
**Major/minor:** Major
**Error location 1:** "inspection wells" instead of "inspection shafts"
**Explanation for error 1:** The term "inspection wells" is not the same as "inspection shafts" and may confuse the reader.
**Error type 2:** Incorrect translation includes information not present in the correct translation
**Major/minor:** Major
**Error location 2:** "inspection wells" instead of "inspection shafts"
**Explanation for error 2:** The addition of "convex" changes the meaning of the structure, which is not present in the correct translation.
**Error type 3:** Incorrect translation is missing content from the correct translation
**Major/minor:** Major
**Error location 3:** "combined masonry" instead of "combined masonry work"
**Explanation for error 3:** The phrase "combined masonry work" is missing, which changes the meaning of the sentence.
**Error type 4:** Incorrect translation has stylistic problems
**Major/minor:** Minor
**Error location 4:** "The advantage of using concrete modules to build inspection wells is that the upper, lower, left and right sides of the modules are all concave-convex groove structure,"
**Explanation for error 4:** The sentence structure is awkward and not as clear as the correct translation.

Table 14: This is the case where error locations are not consistent with the explanation (**M2**).

**INSTRUCTSCORE Input:**
**Reference:** 1. [Baijiu Distillery Direct Sales] Ningcheng Balihan Aged Cellar 42 Degrees 500 ML, Rich Flavor Sorghum Original Baijiu, Aged Spirit Tasting Mellow, Four Bottles in One Box
**Output:** 1. [Directly run by winery] Ningcheng Balihan Laojiao 42 degree 500ML lush-flavored sorghum puree liquor with a soft and refreshing taste, four bottles in the whole box

**INSTRUCTSCORE Output:**
**Your Translation contains 5 errors:**
**Error type 1:** Incorrect translation does not accurately represent the correct translation
**Major/minor:** Major
**Error location 1:** "winery" instead of "distillery"
**Explanation for error 1:** The translation incorrectly states that the business is a winery instead of a distillery, which changes the meaning of the sentence.
**Error type 2:** Incorrect translation includes information not present in the correct translation
**Major/minor:** Major
**Error location 2:** "lush-flavored" instead of "rich-flavored"
**Explanation for error 2:** The translation adds the word "lush," which is not present in the correct translation and changes the meaning of the description.
**Error type 3:** Incorrect translation is missing content from the correct translation
**Major/minor:** Major
**Error location 3:** "aged spirit" instead of "aged baijiu"
**Explanation for error 3:** The translation omits the specific type of alcohol being described, which is important for understanding the product being sold.
**Error type 4:** Incorrect translation has stylistic problems
**Major/minor:** Major
**Error location 4:** "whole box" instead of "box" Explanation for error 4: The translation uses the phrase "whole box," which is not a standard way to describe a box and may confuse the reader.
**Error type 5:** Incorrect translation does not accurately represent the correct translation
**Major/minor:** Major
**Error location 5:** "soft and refreshing taste" instead of "smooth and refreshing taste"
**Explanation for error 5:** The translation changes the adjective "smooth" to "soft," which alters the description of the taste of the alcohol.

Table 15: This is the case where error locations are not referred in the output text (**M3**). "aged spirit" is not never mentioned in the output text.

INSTRUCTSCORE Input:
**Reference:** It is understood that the activities held during the "Children's Day" by China Children's Center this year include the "Cool Maker Loves Family" carnival of children's future learning, the first children's visual art festival, the "Book Carnival" event and the "Whole Family Loves Reading" Picture Book Reading activity.
**Output:** It is understood that the activities held by the China children's center during this year's "June 1" children's day include the "love family and create cool" children's future learning carnival, the first children's visual arts festival, the "book carnival" activity, the "family love reading" picture book series activity, etc.

INSTRUCTSCORE Output:
**Your Translation contains 5 errors:**
**Error type 1:** Incorrect translation includes information not present in the correct translation
**Major/minor:** Major
**Error location 1:** "June 1"
**Explanation for error 1:** The incorrect translation specifies the date of Children's Day as "June 1," while the correct translation says "Children's Day," which does not specify the date.
**Error type 2:** Incorrect translation has stylistic problems
**Major/minor:** Major
**Error location 2:** "love family and create cool"
**Explanation for error 2:** The phrase "love family and create cool" is awkward and does not convey the intended meaning of "celebrating families and creativity."
**Error type 3:** Incorrect translation is missing content from the correct translation
**Major/minor:** Major
**Error location 3:** "Children's Day"
**Explanation for error 3:** The incorrect translation omits the term "Children's Day," which is an important aspect of the correct translation.
**Error type 4:** Incorrect translation does not accurately represent the correct translation
**Major/minor:** Major
**Error location 4:** "picture book series activity"
**Explanation for error 4:** The incorrect translation changes the meaning of the "picture book reading activity" to imply a series of books rather than a single event.
**Error type 5:** Incorrect translation includes information not present in the correct translation
**Major/minor:** Major
**Error location 5:** "whole family loves reading"
**Explanation for error 5:** The incorrect translation adds the phrase "whole family loves reading," which is not present in the correct translation and changes the focus from "families" to "whole families."

Table 16: This is the case where error locations in explanation can not refer to the output text (**M4**). "whole family loves reading" is never mentioned in the output text. Instead, it was mentioned in the reference text.

INSTRUCTSCORE Input:
**Reference:** Therefore, Zhang Zhongmou thinks it is ironic for Intel to work on wafer foundry.
**Output:** So, Zhang Zhongmou feels ironic about Intel's entry into wafer foundry.

INSTRUCTSCORE Output:
**Your Translation contains 2 errors**:
**Error type 1:** Wrong grammatical register (eg, inappropriately informal pronouns). Internal inconsistency (not related to terminology)
**Major/minor:** Major
**Error location 1:** "So"
**Explanation for error 1:** The use of "So" is informal and inconsistent with the formal tone of the rest of the sentence.
**Error type 2:** Incorrect translation is missing content from the correct translation
**Major/minor:** Minor
**Error location 2:** "entry" instead of "entry into"
**Explanation for error 2:** The incorrect translation omits the word "into," which is necessary to convey the meaning of Intel's entry into the wafer foundry market.

Table 17: This is the case where major and minor labels do not correspond to the correct severity levels (**M5**). The use of "So" is not a severe error.

**INSTRUCTSCORE Input**:
**Reference:** Washing hands with air, diagnosing disease through breath, and accurately seeking for source of smog; last night, a series of creative academic achievements were awarded at the opening ceremony of the 2016 Annual Academic Conference of Zhejiang University.
**Output:** Wash hands with air, judge diseases, accurately search for the source of fog Yesterday evening, at the opening ceremony of the 2016 academic annual meeting of Zhejiang University, a series of academic achievements were recognized.

---

**INSTRUCTSCORE Output:**
**Your Translation contains 3 errors:**
**Error type 1:** Incorrect translation is missing content from the correct translation
**Major/minor:** Major
**Error location 1:** "judge diseases"
**Explanation for error 1:** The incorrect translation is missing the word "diagnose" and changes the meaning from accurately determining the cause of a disease to simply judging diseases.
**Error type 2:** Incorrect translation has stylistic problems
**Major/minor:** Major
**Error location 2:** "accurately search for the source of fog"
**Explanation for error 2:** The incorrect translation uses "fog" instead of "smog," which changes the meaning and makes the sentence less clear.
**Error type 3:** Incorrect translation includes information not present in the correct translation
**Major/minor:** Minor
**Error location 3:** "academic annual meeting"
**Explanation for error 3:** The incorrect translation adds the word "annual" to the phrase "academic meeting," which is not present in the correct translation. However, this does not significantly change the meaning of the sentence.

Table 18: This is the case where the explanation is wrong. However, error at a specified location does exist (**M6**). "academic annual meeting" is a minor error. However, explanation is wrong because the word "annual" is also present in the reference.

---

**INSTRUCTSCORE Input**:
**Reference:** This set of biological sensor uses the new material of graphene; its target is chemical elements, and its keen "sense of smell" enables it to reflect the health status of the body more in-depth and accurately.
**Output:** This biosensor uses graphene, a new material, and its target is chemical elements. Its keen sense of smell enables it to reflect the health status more deeply and accurately.

---

**INSTRUCTSCORE Output:**
**Your Translation contains 2 errors:**
**Error type 1:** Incorrect translation is missing content from the correct translation
**Major/minor:** Major
**Error location 1:** "and its target is chemical elements"
**Explanation for error 1:** The incorrect translation omits the information about the biosensor's target being "chemical elements" and instead only mentions "chemical elements."
**Error type 2:** Incorrect translation has stylistic problems
**Major/minor:** Minor
**Error location 2:** "biosensor" instead of "biological sensor"
**Explanation for error 2:** The incorrect translation uses "biosensor" instead of "biological sensor," which is a minor stylistic error that does not significantly change the meaning but may be noticed by the reader.

Table 19: Error described in the explanation is not an error (**G1**). The incorrect phrase and suggested revision are exactly the same.

**INSTRUCTSCORE Input**:
**Reference**: And how much longer does it take for the restaurant to prepare it? Could could help me ask them?
**Output:** Or how long before the restaurant is ready? Can you ask for me?

**INSTRUCTSCORE Output:**
**Your Translation contains 3 errors:**
**Error type 1:** Incorrect translation is missing content from the correct translation
**Major/minor:** Major
**Error location 1:** "Or how long before the restaurant is ready?"
**Explanation for error 1:** The incorrect translation omits the information about the wait time for the food, which is an important detail in the correct translation.
**Error type 2:** Incorrect translation has stylistic problems
**Major/minor:** Major
**Error location 2:** "Can you ask for me?"
**Explanation for error 2:** The incorrect translation changes the meaning of the sentence by implying that the person is asking the question on behalf of someone else, rather than themselves.
**Error type 3:** Problems with grammar, other than orthography
**Major/minor:** Minor
**Error location 3:** "could help me ask them?"
**Explanation for error 3:** The incorrect translation uses "could help me ask them" instead of "could you ask them for me?" The grammar is slightly off, but the meaning is still clear.

Table 20: One error is mentioned more than once among explanations (**G2**).

**INSTRUCTSCORE Input**:
**Reference:** Normally, people wash hands by flushing hands with water, with a little soap and sanitizer at most; however, Li Qizhang and his classmates invented a set of "equipment to wash hands with air".
**Output:** Ordinary people wash their hands with water, at most, wipe some soap and squeeze some hand sanitizer, while Li Qizhang and other students invented a set of "air hand washing device".

**INSTRUCTSCORE Output:**
**Your Translation contains 4 errors:**
**Error type 1:** Incorrect translation is missing content from the correct translation
**Major/minor:** Major
**Error location 1:** "with air"
**Explanation for error 1:** The incorrect translation changes the method of hand washing from "with water" to "with air," which significantly alters the meaning of the sentence.
**Error type 2:** Incorrect translation has stylistic problems
Major/minor: Major
**Error location 2:** "Ordinary people wash their hands with water, at most, wipe some soap and squeeze some hand sanitizer,"
**Explanation for error 2:** The incorrect translation has a choppy and awkward sentence structure, making it difficult to understand the intended meaning.
**Error type 3:** Incorrect translation includes information not present in the correct translation
Major/minor: Major
**Error location 3:** "other students"
**Explanation for error 3:** The incorrect translation adds information about other students, which is not present in the correct translation.
**Error type 4:** Incorrect translation does not accurately represent the correct translation
**Major/minor:** Minor
**Error location 4:** "air hand washing device"
**Explanation for error 4:** The incorrect translation uses "air hand washing device" instead of "air hand washing equipment," which is a less accurate representation of the correct translation.

Table 21: This is the case where two phrases are misaligned in the explanation (**G3**). "With water" and "With air" are incorrectly aligned. "with air" is the correct generation output and is not an error.

**INSTRUCTSCORE Input**:
**Reference:** Wang Cheng pointed out that hosting the talent summit was an important measure to implement the spirit of the talent working conference of the central government and the spirit of the 12th Party Congress of Hunan, showing the sincerity of Hunan in terms of putting emphasis on talents, being eager for talents and making full use of talents.
**Output:** Wang Cheng pointed out that the talent summit is an important initiative to implement the central talent work conference and the spirit of the 12th Party Congress of the province, which shows the sincere sincerity of Hunan in attaching importance to talents, thirsting for talents and using talents well.

**INSTRUCTSCORE Output:**
**Your Translation contains 2 errors:**
**Error type 1:** Incorrect translation does not accurately represent the correct translation
**Major/minor:** Major
**Error location 1:** "the central talent work conference" and "the 12th Party Congress of the province"
**Explanation for error 1:** The incorrect translation changes the meaning by referring to different events and locations than the correct translation.
**Error type 2:** Incorrect translation has stylistic problems
**Major/minor:** Minor
**Error location 2:** "thirsting for talents"
**Explanation for error 2:** The phrase "thirsting for talents" is awkward and not commonly used, which makes it a stylistic issue.

Table 22: One error span mentions multiple errors (**G4**)

---

**INSTRUCTSCORE Input**:
**Reference:** Washing hands with air, diagnosing disease through breath, and accurately seeking for source of smog; last night, a series of creative academic achievements were awarded at the opening ceremony of the 2016 Annual Academic Conference of Zhejiang University.
**Output:** Wash hands with air, judge diseases, accurately search for the source of fog Yesterday evening, at the opening ceremony of the 2016 academic annual meeting of Zhejiang University, a series of academic achievements were recognized.

**INSTRUCTSCORE before refinement's output:**
Your Translation contains 3 errors:
Error type 1: Incorrect translation is missing content from the correct translation
Major/minor: Major
Error location 1: "judge diseases" Explanation for error 1: The incorrect translation is missing the word "diagnose" and changes the meaning from accurately determining the cause of a disease to simply judging diseases.
Error type 2: Incorrect translation has stylistic problems
Major/minor: Major
Error location 2: "accurately search for the source of fog"
Explanation for error 2: The incorrect translation uses "fog" instead of "smog," which changes the meaning and makes the sentence less clear.
Error type 3: Incorrect translation includes information not present in the correct translation
Major/minor: Minor
Error location 3: "academic annual meeting"
Explanation for error 3: The incorrect translation adds the word "annual" to the phrase "academic meeting," which is not present in the correct translation. However, this does not significantly change the meaning of the sentence.

**INSTRUCTSCORE after refinement's output:**
Your Translation contains 3 errors:
Error type 1: Incorrect translation is missing content from the correct translation
Major/minor: Major
Error location 1: "judge diseases" Explanation for error 1: The incorrect translation is missing the information about "diagnosing diseases" from the correct translation.
Error type 2: Incorrect translation has stylistic problems
Major/minor: Major
Error location 2: "accurately search for the source of fog"
Explanation for error 2: The incorrect translation uses "fog" instead of "smog," which changes the meaning and makes it less clear.
Error type 3: Incorrect translation does not accurately represent the correct translation
Major/minor: Minor
Error location 3: "academic annual meeting"
Explanation for error 3: The incorrect translation uses "academic annual meeting" instead of "academic annual conference," which is a minor error that does not significantly change the meaning but will be noticed.

Table 23: In this case, INSTRUCTSCORE before refinement is able to correctly identify the error location, type and major/minor label. However, its explanation over the error is incorrect. INSTRUCTSCORE after refinement improves model's rationable and output the correct explanation.

**INSTRUCTSCORE Input:**
**Reference:**
Diese Vakuumpumpe mit Mehrfachbetrieb hat einen speziellen, in den Pumpenkopf integrierten Griff, der dem Benutzer eine exzellente Kontrolle über den Pump- und Saugvorgang gibt."
**Output:**
Diese mehrfach angetriebene Vakuumpumpe verfügt über einen speziellen Griff, der in den Pumpenkopf eingebaut ist und dem Benutzer eine hervorragende Kontrolle über den Pump- und Saugvorgang gibt.

**INSTRUCTSCORE Output:**
**Your Translation contains 2 errors:**
**Error type 1:** Incorrect translation is missing content from the correct translation
**Major/minor:** Major
**Error location 1:** "mit Mehrfachbetrieb" is missing in the incorrect translation
**Explanation for error 1:** The incorrect translation omits the important information that the vacuum pump is multifunctional, which changes the meaning of the sentence.
**Error type 2:** Incorrect translation has stylistic problems
**Major/minor: Minor**
**Error location 2:** "hervorragende" instead of "exzellente"
**Explanation for error 2:** The use of "hervorragende" is not incorrect, but it is less commonly used in this context and may be considered a stylistic choice. The meaning is still clear, but it may be noticed by the reader

Table 24: Candidate and reference texts are both in German. However, instruction and diagonostic report are in English to enhance user's understandings. From this example, we can see the generation of the Non-English diagnostic report requires mixed code output and multilingual understandings between English and target language.

**Prompt**: Find 100 major domains at machine translation following format like this: News, Technical, Legal, Medical, Financial, Gaming, E-commerce, Tourism and hospitality, Marketing and advertising, and Scientific (make sure all domains are very distinct from one another)

**GPT-4:** "News","Technical","Legal","Medical","Financial","Gaming","E-commerce","Tourism and hospitality", "Marketing and advertising","Scientific","Education","Social media","Entertainment","Automotive","Sports","Retail", "Energy","Agriculture","Aerospace","Art and culture","Beauty and personal care","Construction","Consumer goods", "Defense","Electronics","Environment","Food and beverage","Government","Health and fitness","Insurance","Internet of things (IoT)","Logistics","Manufacturing","Music","Non-profit","Pharmaceuticals","Real estate","Recruitment","Religion","Telecommunications","Transportation","Utilities","Venture capital","Virtual and augmented reality","Web development","Wine and spirits","Women's issues","Workplace and HR","Youth issues","Zoology","Robotics and automation","Urban planning and development","Parenting and family","Mental health and wellness","Biotechnology","Cybersecurity","Film and television","Language and linguistics","Photography and videography","Politics and diplomacy","Psychology and neuroscience","Public health and safety","Publishing and journalism","Radio and podcasts","Renewable energy and sustainability","Research and development","Shipping and maritime","Space and astronomy","Start-ups and entrepreneurship", "Taxation and accounting","Textiles and fashion","Theatre and performing arts","Travel and exploration","Weather and meteorology","Wildlife and conservation","Animation and graphic design","Artificial intelligence and machine learning","Blockchain and cryptocurrency","Cannabis and hemp industries","Chemical engineering and materials science","Coaching and personal development","Comic books and graphic novels", "Dance and choreography","Diet and nutrition","Digital marketing and SEO","Disaster relief and emergency management","E-learning and online courses","Event planning and management","Forensic science and criminology","Genealogy and family history","Geology and earth sciences","Home and garden","Industrial design and product design","Intellectual property and patents","Interior design and architecture","Library and information science", "Marine biology and oceanography","Meteorology and climatology","Mining and minerals","Pet care and animal husbandry"

Table 25: This is one example of how we generate 100 distinct domains from GPT-4.

| **Prompt**: Find 100 topics in the Technical domain (make sure all topics are very distinct from one another) |
|---|
| **GPT-4:** 1. Artificial Intelligence 2. Machine Learning 3. Data Science 4. Natural Language Processing 5. Robotics 6. Internet of Things 7. Big Data 8. Cyber Security 9. Cloud Computing 10. Virtual Reality 11. Augmented Reality 12. Blockchain 13. Cryptocurrencies 14. Quantum Computing 15. 5G Technology 16. Wireless Communication 17. Computer Vision 18. Neural Networks 19. Deep Learning 20. Human-Computer Interaction 21. User Experience Design 22. Software Development 23. Web Development 24. Mobile App Development 25. Game Development 26. Computer Graphics 27. Computer Animation 28. Computer Networks 29. Operating Systems 30. Database Management 31. Information Retrieval 32. Computer Architecture 33. Microprocessors 34. Embedded Systems 35. Control Systems 36. Mechatronics 37. Electrical Engineering 38. Electronic Engineering 39. Mechanical Engineering 40. Civil Engineering 41. Chemical Engineering 42. Materials Science 43. Nanotechnology 44. Biomedical Engineering 45. Bioinformatics 46. Computational Biology 47. Genetics 48. Neuroscience 49. Psychology 50. Linguistics 51. Computational Linguistics 52. Digital Signal Processing 53. Image Processing 54. Audio Processing 55. Video Processing 56. Speech Recognition 57. Speech Synthesis 58. Text-to-Speech 59. Optical Character Recognition 60. Computer Vision Applications 61. Face Recognition 62. Object Detection 63. Object Tracking 64. Image Segmentation 65. Image Classification 66. Image Retrieval 67. Image Compression 68. Video Compression 69. Video Streaming 70. Multimedia Systems 71. Human-Robot Interaction 72. Human-Centered Computing 73. Ubiquitous Computing 74. Wearable Computing 75. Smart Homes 76. Smart Cities 77. Intelligent Transportation Systems 78. Autonomous Vehicles 79. Drones 80. Industrial Automation 81. Manufacturing Systems 82. Quality Control 83. Supply Chain Management 84. Logistics 85. Health Informatics 86. Telemedicine 87. Electronic Health Records 88. Medical Imaging 89. Medical Devices 90. Biomechanics 91. Rehabilitation Engineering 92. Assistive Technology 93. Education Technology 94. E-Learning 95. Learning Analytics 96. Gamification 97. Social Media Analytics 98. Web Analytics 99. Search Engine Optimization 100. Digital Marketing. |

Table 26: This is one example of how we generate 100 topics within a certain domain. Here are 100 topics generated by GPT-4 for the Technical domain.

| **Prompt**: Give me 5 English sentences about 5G Technology (Each has a distinct topic, length, and sentence structure). Only return English text. |
|---|
| **GPT-4:** |
| 1. With the introduction of 5G technology, we can expect significantly faster download speeds, enabling quicker access to high-quality content such as 4K videos and virtual reality experiences. |
| 2. One of the most exciting aspects of 5G technology is its potential for improving communication between smart devices in the Internet of Things, paving the way for a more connected and efficient society. |
| 3. Enhanced safety features and real-time data transfer in 5G-powered autonomous vehicles will revolutionize the way we commute, offering faster, greener, and more reliable transportation. |
| 4. 5G networks are anticipated to make dramatic improvements in telemedicine; for example, doctors could conduct remote consultations and perform surgeries with haptic feedback, ultimately increasing accessibility to healthcare. |
| 5. Although there are concerns surrounding the potential health risks and privacy issues associated with 5G technology, many experts argue that the benefits far outweigh the drawbacks, heralding a new era of innovation and connectivity. |

Table 27: This is one example how we generate 5 distinct sentences for distinct sentence length and structure from GPT-4 for a given topic.

| Error Category | | Description |
|---|---|---|
| Accuracy | Addition | Translation includes information not present in the source. |
| | Omission | Translation is missing content from the source. |
| | Mistranslation | Translation does not accurately represent the source. |
| | Untranslated text | Source text has been left untranslated. |
| Fluency | Spelling | Incorrect spelling or capitalization. |
| | Grammar | Problems with grammar, other than orthography. |
| | Register | Wrong grammatical register (eg, inappropriately informal pronouns). |
| | Inconsistency | Internal inconsistency (not related to terminology). |
| | Character encoding | Characters are garbled due to incorrect encoding. |
| Terminology | Inappropriate for context | Terminology is non-standard or does not fit the context. |
| | Inconsistent use | Terminology is used inconsistently. |
| Style | Awkward | Translation has stylistic problems. |
| Locale convention | Address format | Wrong format for addresses. |
| | Currency format | Wrong format for currency. |
| | Date format | Wrong format for dates. |
| | Name format | Wrong format for names. |
| | Telephone format | Wrong format for telephone numbers. |
| | Time format | Wrong format for time expressions. |

Table 28: We use the above error types to perform data synthesis at GPT-4. These definitions are from Freitag et al. (2021a). For each error annotation, we randomly select one of the above error types and query GPT4 for the generation. Please check the prompt example at Table 29.

---

**Prompt**: You are evaluating a model output based on a reference, which describes semantics about text. The correct translation is, "*The art of writing for educational publications involves striking a delicate balance between providing enough detail to be useful and overwhelming the reader with too much information.*". Your translation needs to contain two non-overlapping errors non-overlapping errors based on the given error types. Please give 0 minor errors and 2 major errors. Major errors can significantly alter the meaning of the sentences, while minor errors will not change the meaning. For each error, give me an error type, a major/minor label, an error location, and an explanation. The format is Paraphrase correct translation:
Incorrect Translation:
Error type 1: Translation includes information not present in the correct translation
Major/minor: Major
Error location 1:
Explanation for error 1:
Error type 2: Problems with grammar, other than orthography
Major/minor: Major
Error location 2:
Explanation for error 2:

---

**GPT-4:** Paraphrase correct translation: "*The art of writing for educational publications involves striking a delicate balance between providing enough detail to be useful and overwhelming the reader with too much information.*"
Incorrect Translation: "*Writing for educational publications is about finding a balance between giving enough examples to be useful and making the reader bored with too many details.*"
Error type 1: Translation includes information not present in the correct translation
Major/minor: Major
Error location 1: "...giving enough examples..."
Explanation for error 1: The correct translation talks about providing enough detail, not specific examples, which changes the meaning of the sentence.
Error type 2: Problems with grammar, other than orthography
Major/minor: Major
Error location 2: "...making the reader bored with too many details."
Explanation for error 2: The correct translation mentions overwhelming the reader with too much information, not making them bored with too many details. This change in phrasing alters the meaning of the sentence.

---

Table 29: This prompt was used to generate synthetic data to fine-tune the LLAMA model, specifically GPT-4. The model was provided with the number of errors in a sentence. For each error annotation, we pre-define the error type and a severity label. GPT-4 was then required to identify the error locations and provide explanations for each error annotation. The prompts in red are variables for each synthesized generation.

**Prompt**: You are evaluating a model output based on a reference, which describes semantics about another modality, such as image. The correct generation is, "A very large kite is laying on the beach.". Your generation needs to contain two non-overlapping errors based on the given error types. Please give 2 minor errors and 0 major error. Major errors can confuse or mislead the reader due to significant change in meaning, while minor errors don't lead to loss of meaning but errors are still noticeable. For each error, give me an error type, a major/minor label, an error location and an explanation. If some error types with major/minor labels do not exist at incorrect generation, put N/A at error location and explanation. The format is
Paraphrased correct generation:
Incorrect Generation:
Error type 1: The caption wrongly describes motion or movement in the correct generation
Major/minor: Minor
Error location 1:
Explanation for error 1:
Error type 2: The caption wrongly describes object relationships in the correct generation
Major/minor: Minor
Error location 2:
Explanation for error 2:

---

**GPT-4:** Paraphrased correct generation: A huge kite rests on the sandy beach.
Incorrect Generation: A huge kite is flying towards the sandy beach.
Error type 1: The caption wrongly describes motion or movement in the correct generation
Major/minor: Minor
Error location 1: "flying towards"
Explanation for error 1: The kite is described as laying on the beach in the correct generation, while in the incorrect generation it is described as flying towards the beach.
Error type 2: The caption wrongly describes object relationships in the correct generation
Major/minor: Minor
Error location 2: "towards"
Explanation for error 2: The incorrect generation suggests that the kite is moving towards the beach, while in the correct generation, the kite is already on the beach. This changes the relationship between the objects (kite and beach).

Table 30: This prompt was used to generate synthetic data to fine-tune the LLAMA model, specifically GPT-4. The model was provided with the number of errors in a sentence. For each error annotation, we pre-define the error type and a severity label. GPT-4 was then required to identify the error locations and provide explanations for each error annotation. The prompts in green are variables for each synthesized generation.

---

**Prompt**: You are evaluating a model output based on commonsense knowledge. Use "enter, exit, shopper, store" to write a sentence which follows common sense. One example generation is, "shoppers enter and exit a store .". Your generation needs to contain two non-overlapping errors based on the given error types. Please give 0 minor error and 2 major errors. Major errors can confuse or mislead the reader due to significant change in meaning, while minor errors don't lead to loss of meaning but errors are still noticeable. For each error, give me an error type, a major/minor label, an error location and an explanation. If some error types with major/minor labels do not exist at incorrect generation, put N/A at error location and explanation. The format is
Paraphrased correct generation:
Incorrect Generation:
Error type 1: Actions/beliefs contradicting known personality
Major/minor: Major
Error location 1:
Explanation for error 1:
Error type 2: Inaccurate characteristics/abilities of objects
Major/minor: Major
Error location 2:
Explanation for error 2:

---

**GPT-4:** Paraphrased correct generation: Customers come in and out of the store.
Incorrect Generation: Shoppers fly in and stay forever in the store.
Error type 1: Inaccurate characteristics/abilities of objects
Major/minor: Major
Error location 1: Shoppers fly
Explanation for error 1: Shoppers cannot fly; they walk or move using other means such as wheelchairs or carts.
Error type 2: Actions/beliefs contradicting known personality
Major/minor: Major
Error location 2: stay forever in the store Explanation for error 2: Shoppers do not stay in the store forever; they eventually leave after making their purchases or browsing.

Table 31: This prompt was used to generate synthetic data to fine-tune the LLAMA model, specifically GPT-4. The model was provided with the number of errors in a sentence. For each error annotation, we pre-define the error type and a severity label. GPT-4 was then required to identify the error locations and provide explanations for each error annotation. The prompts in green are variables for each synthesized generation.

**Prompt**: You are evaluating a model output based on a structured data input and a reference. The input is '1634:The Bavarian Crisis | author | Eric Flint', '1634: The Bavarian Crisis | mediaType | Hardcover', '1634: The Bavarian Crisis | numberOfPages | "448"'. The correct generation is, "1634: The Bavarian Crisis, written by Eric Flint, has 448 pages and is available in hardcover.". Your generation needs to contain one non-overlapping error based on the given error types. Please give 1 minor error and 0 major error. Major errors can confuse or mislead the reader due to significant change in meaning, while minor errors don't lead to loss of meaning but errors are still noticeable. For each error, give me an error type, a major/minor label, an error location and an explanation. If some error types with major/minor labels do not exist at incorrect generation, put N/A at error location and explanation.
The format is:
Paraphrased correct generation:
Incorrect Generation:
Error dimension 1: Data Coverage
Error type 1: The text only partially describes a predicate present in the data
Major/minor: Minor
Error location 1:
Explanation for error 1:

**GPT-4:** Paraphrased correct generation: "1634: The Bavarian Crisis, written by Eric Flint, has 448 pages and is available in hardcover."
Incorrect Generation: "1634: The Bavarian Crisis, written by Eric Flint, is available in hardcover."
Error dimension 1: Data Coverage
Error type 1: The text only partially describes a predicate present in the data
Major/minor: Minor
Error location 1: author and numberOfPages
Explanation for error 1: The incorrect generation omits the predicate "numberOfPages", which results in incomplete information and noticeable errors in data coverage.

Table 32: This prompt was used to generate synthetic data to fine-tune the LLAMA model, specifically GPT-4. The model was provided with the number of errors in a sentence. For each error annotation, we pre-define the error type and a severity label. GPT-4 was then required to identify the error locations and provide explanations for each error annotation. The prompts in green are variables for each synthesized generation.

**Prompt**: You are translating from English-to-German. The correct translation is, "Die Anzahl der COVID-19-Fälle in Deutschland hat einen neuen Höchststand erreicht.". First, paraphrase the correct translation. Then, generate a translation that contains four non-overlapping errors based on the given error types. Please give 2 minor errors and 2 major errors. Major errors can confuse or mislead the reader due to significant change in meaning, while minor errors don't lead to loss of meaning but will be noticed. For each error, give me an error type, a major/minor label, an error location and an explanation. If some error types with major/minor labels do not exist at incorrect translation, put N/A at error location and explanation. The format is:
Paraphrased correct translation:
Incorrect Translation:
Error type 1: Terminology is non-standard or does not fit context. Terminology is used inconsistently
Major/minor: Major
Error location 1:
Explanation for error 1:
Error type 2: Incorrect translation includes information not present in the correct translation (Not the case of opposing the meaning from correct translation)
Major/minor: Major
Error location 2:
Explanation for error 2:
Error type 3: Incorrect translation does not accurately represent the correct translation (Not the case of opposing the meaning from correct translation)
Major/minor: Minor
Error location 3:
Explanation for error 3:
Error type 4: Incorrect translation has stylistic problems
Major/minor: Minor
Error location 4:
Explanation for error 4:

---

**GPT-4:**
Paraphrased correct translation: "In Deutschland ist die Zahl der COVID-19-Fälle auf einen neuen Rekord gestiegen."
Incorrect Translation: "Die Summe von COVID-19-Äpfel in Deutschland hat einen neuen Höchstwert erreichen."
Error type 1: Terminology is non-standard or does not fit context. Terminology is used inconsistently
Major/minor: Major
Error location 1: "COVID-19-Äpfel"
Explanation for error 1: The term "COVID-19-Äpfel" is incorrect and does not fit the context. The correct term should be "COVID-19-Fälle".
Error type 2: Incorrect translation includes information not present in the correct translation (Not the case of opposing the meaning from correct translation)
Major/minor: Major
Error location 2: "Summe"
Explanation for error 2: The use of the word "Summe" adds information not present in the correct translation. The correct word should be "Anzahl" or "Zahl".
Error type 3: Incorrect translation does not accurately represent the correct translation (Not the case of opposing the meaning from correct translation)
Major/minor: Minor
Error location 3: "erreichen"
Explanation for error 3: The verb "erreichen" is in the infinitive form, while it should be in the past participle form "erreicht" to represent the correct translation accurately.
Error type 4: Incorrect translation has stylistic problems
Major/minor: Minor
Error location 4: "von"
Explanation for error 4: The preposition "von" is not necessary in this context, and it creates a stylistic problem. The correct translation should be "Die Anzahl der COVID-19-Fälle".

Table 33: This prompt was used to generate synthetic data to fine-tune the LLAMA model, specifically GPT-4. The model was provided with the number of errors in a sentence. For each error annotation, we pre-define the error type and a severity label. GPT-4 was then required to identify the error locations and provide explanations for each error annotation. The prompts in green are variables for each synthesized generation.

---

**Prompt** (For diagnostic report which contains error annotations):
*Ref: State-owned enterprises and advantageous private enterprises entered the revolutionary base area of south Jiangxi.*
*Out: State-owned enterprises and dominant private enterprises entered the old revolutionary district of southern Jiangxi.*
*Error1:*
*Error location 1: "old revolutionary district"*
*Error type 1: Terminology is non-standard or does not fit the context. Terminology is used inconsistently*
*Explanation 1: The correct term should be "new revolutionary base area" which refers to the area where the revolution started in 1927, not "old revolutionary district" which does not convey the same meaning.*
*Error2:*
*Error location 2: "dominant"*
*Error type 2: Incorrect translation includes information not present in the correct translation*
*Explanation 2: The word "dominant" is used instead of "privileged," which adds information that is not present in the correct translation, but does not change the overall meaning.*

Q1: For each error location, extract the incorrect error location.
A1:

Q2: Parse the explanation into either one of the four forms: [incorrect phrase, correct phrase], [incorrect phrase, None], [None, correct phrase], or [None, None].
A2:

Q3: If A2 is "incorrect phrase to correct phrase", is A2 a correct alignment for reference and output? (Answer Yes/No) If A2 contains None, Answer None.
A3:

Q4: According to the sentence context, if A2 is "incorrect phrase to correct phrase" and incorrect and correct phrases can be used interchangeably, it is no-error. If an error can be fixed by reading the rest of the output context it is a minor-error. If it changes the sentence's meaning (Don't consider sentence formality changes) and is unable to be fixed by reading the rest of the output it is a major-error. Answer, no-error or minor-error or major-error
A4:

Q5: Is the explanation consistent with the given error type?
A5:

Q6: Does the explanation talk about the given error location?
A6:

Q7: Do two error locations mention the same location in two different ways? (Answer Yes/No) How many pairs of repetitions? (Answer in a number)
A7: Yes/No, number

The output format will be in JSON
{{Err1: {Q1: A1, Q2: A2, Q3: A3, Q4: A4, Q5: A5, Q6: A6}, Err2: {Q1: A1, Q2: A2, Q3: A3, Q4: A4, Q5: A5, Q6: A6}, Q7: A7}

---

**GPT-4:** { "Err1": { "Q1": "old revolutionary district", "Q2": ["old revolutionary district", "new revolutionary base area"], "Q3": "Yes", "Q4": "major-error", "Q5": "Yes", "Q6": "Yes" }, "Err2": { "Q1": "dominant", "Q2": ["dominant", "privileged"], "Q3": "Yes", "Q4": "minor-error", "Q5": "Yes", "Q6": "Yes" }, "Q7": "No, 0" }

---

Table 34: Human-instructed annotation: if our diagnostic report contains error annotations, we asked following questions to determine the correctness and consistency of the explanation output. This is a prompt example of how we obtained GPT-4 feedback for Table 2.

---

**Prompt** (For diagnostic report which contains no-error):
*Reference: Summarizing foreign media reports, the IAEA report states that Iran's Natanz Nuclear Facility's advanced centrifuge is "accumulating or ready to accumulate enriched uranium".*
*Output: According to comprehensive foreign media reports, the IAEA report pointed out that the advanced centrifuges at Iran's Natanz nuclear facility are "accumulating or are ready to accumulate enriched uranium".*
Compared to the reference, does the output contain any error? (Answer in Yes/No)

---

**GPT-4:** No

---

Table 35: Human-instructed annotation: if our diagnostic report does not contain error annotation, we directly query GPT-4 to validate this claim. If GPT-4 reconfirms with INSTRUCTSCORE's claim, the feedback score is 1. Otherwise, 0.