# OpenReview forum: "INSTRUCTSCORE: Towards Explainable Text Generation Evaluation with Automatic Feedback"
_EMNLP/2023/Conference — EMNLP 2023 Main_

### Official Review · Reviewer_k7RG · 2023-07-19

**Typos Grammar Style And Presentation Improvements:** N/A
**Soundness:** 4

**Excitement:**

3: Ambivalent: It has merits (e.g., it reports state-of-the-art results, the idea is nice), but there are key weaknesses (e.g., it describes incremental work), and it can significantly benefit from another round of revision. However, I won't object to accepting it if my co-reviewers champion it.

**Missing References:**

N/A

**Paper Topic And Main Contributions:**

This paper proposes to fine-tune LLMs for explainable scoring on generation tasks. The correlation with human judgments is on-par with SOTA models and provides additional explanations on scoring for generation tasks.

**Questions For The Authors:**

Can the method generalize to NLU tasks?

**Reasons To Accept:**

The results are good and make a good fit into the current trend on LLMs. This is good research with LLMs and evaluations on generation tasks.

**Reasons To Reject:**

The experiments on MT consider Zh-En only.

**Reproducibility:**

3: Could reproduce the results with some difficulty. The settings of parameters are underspecified or subjectively determined; the training/evaluation data are not widely available.

**Reviewer Confidence:**

4: Quite sure. I tried to check the important points carefully. It's unlikely, though conceivable, that I missed something that should affect my ratings.

---

> ### Author Rebuttal · Authors · 2023-08-28
>
> Thanks for your review and suggestions.
>
> Q.A: English to German experiment
>
> We also run experiments with InstructScore for English-to-German (see Figure 6). InstructScore outperforms all unsupervised with learned metrics except GPT3-based GEMBA metric.
>
> Q.B: Extension to NLU
>
> Applying InstructScore to NLU is an interesting research direction. However, NLU mostly uses classification-based metrics, such as accuracy. The explanation for NLU tasks tend to focus on explaining the model behaviors after making predictions. Different from this work, our explanation focuses on explaining the evaluation process.

---

### Official Review · Reviewer_p5F9 · 2023-07-31

**Soundness:** 4

**Excitement:**

3: Ambivalent: It has merits (e.g., it reports state-of-the-art results, the idea is nice), but there are key weaknesses (e.g., it describes incremental work), and it can significantly benefit from another round of revision. However, I won't object to accepting it if my co-reviewers champion it.

**Paper Topic And Main Contributions:**


The paper proposes a new automatic metric with an error analysis and explanation capability for text generation tasks. The proposed metric is based on Llama-7B as a backbone, which is further finetuned on pseudo-generation outputs (synthetic data from GPT4). After a finetuning stage, the proposed InstructScore is further refined by GPT4 feedback to reduce the risks or failure modes in smaller LLM (Llama-7B) due to its constrained generalization capability compared to when utilizing larger LLMs.

The paper demonstrates that the proposed InstructScore is superior on seen tasks: Machine Translation (WMT22), Wikipedia triples-based text generation (WebNLG20), binary classification of image captioning (Flicker3K-CF) and common sense reasoning (CommonGen); and unseen task: keyword-based text generation (Bagel); compared to currently available automatic evaluation metrics in NLG. The evaluation is done by comparing the correlation level between automatic metrics and human evaluation outcomes.

The contribution of the paper is NLG engineering experiment with an extended analysis (failure mode analysis, ablation study of human evaluation before and after the refinement of the proposed InstructScore).



**Reasons To Accept:**

-	The paper further utilizes LLM’s feedback as a syntetic data (human-like feedback) to improve NLG evaluation with human-like reasoning and error analysis.

-	The authors discuss failure modes of explanation-based evaluation and provide solutions through the proposed metric.

Although, I still have a concern for relying on GPT4’s feedback to meta-evaluate the provided list of failure modes (Table 1). But, if such feedback will be made publicly available as meta-evaluation data for future studies, then at least the authors help advancing current limitations in NLG evaluation field by sharing the data publicly.

-	The paper demonstrates the successful utilization of the proposed evaluation metrics on smaller LLM (7B model compared to larger LLMs, such that 175B GPT3 and GPT4).

-	The by-products: synthetic finetuning data and LLM’s feedback can be useful for future studies.

-	The implication of the study can be expanded to tackle the subjectivity issue of human evaluation in NLG. For example, to increase the objectivity in human evaluation setup by simplifying the task to binary classification of error analysis.


**Reasons To Reject:**


-	The proposed InstructScore is highly dependent on the synthetic training data from OpenAI GPT4 (to finetune LLAMA or the backbone of InstructScore) and GPT4’s feedback (to further finetune LLAMA given its failure modes).

-	The utilization of the proposed automatic metric may be constrained by the API’s particular usage permission.

Could the authors double-check whether the OpenAI license permission (restriction) allows users to use the outputs based on their API to compare with their models (GPT3 and GPT4)? –

This issue might not be the biggest concern if future studies are not using the proposed metric to evaluate GPT3 or GPT4 (or any OpenAI LLMs).

-	Some comparison is rather unfair and over-claimed.

Line 076: Comparison with BLEURT.
BLEURT scores are known to be noisy, which can be linked to the model capacity (BERT, a Transformers MLM model with less number of parameters and generalization capability – to be compared to LLMs with 7B parameter size in the current proposal). Even though BLEURT is supervisedly trained or aligned with constrained human ratings, I would not be surprised that the proposed LLM-based metric outperforms BLEURT, considering the benefit of scaling (model size and pretraining data) from proprietary API-based LLMs.

The metric evaluation (in Table 2) is also rather apple-to-orange comparison. Most of the metrics, such as BLEURT, BERTScore, BARTScore, etc are not automatic metrics that are optimized to reason the model performance with an error analysis. For each metric, the correlation level to the human evaluators is calculated by a different setup. For example, human ratings is often done by voting based on relative comparison, while in InstructScore, human is asked to focus more on binary classification of each error type.

I would be more interested if in addition to the final correlation score level between automatic metrics and human raters, the authors also highlight the increase/decrease of human agreement level given the different human evaluation setup above (I see that in Appendix line 991, the agreement level between three raters is defined as 0.388). I would have an educational guess that supporting human evaluation with an error analysis objective may increase the agreement level between raters because the evaluators are becoming more objective, compared to when they are subjectively comparing between two different generation outputs based on predefined rating scale.




**Reproducibility:**

4: Could mostly reproduce the results, but there may be some variation because of sample variance or minor variations in their interpretation of the protocol or method.

**Reviewer Confidence:**

4: Quite sure. I tried to check the important points carefully. It's unlikely, though conceivable, that I missed something that should affect my ratings.

---

> ### Author Rebuttal · Authors · 2023-08-28
>
> Thanks so much for your review of our work.
>
> Q.A: OpenAI restrictions:
>
> OpenAI’s terms state that “OpenAI prohibits using their services to develop models that compete with OpenAI”. Our InstructScore does not aim to compete against OpenAI in any of their products or redistribute GPT4’s output data. Rather, our objective is to provide an open-sourced and explainable evaluation metric and explore LLMs for fine-grained categorization of errors. Therefore it is acceptable under their terms.
>
> Q.B: Comparison set up:
>
> We agree that model size and pre-training data plays an important role for the performance of the learned evaluation metric. However, the main focus of the paper is to propose a novel framework of explainable evaluation metric rather than extensively examining the scaling properties. We evaluate this metric in two settings: 1) its final score’s correlation to the human ratings 2) human evaluation at its diagnostic output. For the first setting, we include BLEURT and COMET as comparison points, which indicates InstructScore’s correlation compared to SOTA metrics at each task. However, we do not claim InstructScore can outperform BLEURT and COMET under the same training data and model size. For the second setting, we leverage fine-grained human annotation to meta-evaluate InstructScore’s outputs. We further include InstructScore’s precision and recall of its annotations and explanations to illustrate its usefulness to humans. Our human evaluation results indicate that, after refinement, 89.4% of InstructScore's output fields are correct, and it can identify 85.6% of error fields produced by the translation system. Furthermore, regarding explanations, 86.1% of InstructScore's output after refinement is deemed correct by humans, and it can explain 85.0% of errors produced by the translation system. We will revise our claim in the paper.
>
> Q.C: InstructScore to assist human evaluation:
>
> Thanks for your great suggestion. We believe using InstructScore to support human labeling is a great idea and future work.  This would exceed the scope of this work and would require more additional budget to run human evaluation with professional translators (the standard in MT).
>
> | All fields | Before refinement | After refinement |
> | ---------- | ----------------- | ---------------- |
> | Precision  | 0.778             | 0.894            |
> | Recall     | 0.824             | 0.856            |
>
> | Explanation field | Before refinement | After refinement |
> | ----------------- | ----------------- | ---------------- |
> | Precision         | 0.756             | 0.861            |
> | Recall            | 0.819             | 0.850             |

---

### Official Review · Reviewer_uTpq · 2023-08-05

**Soundness:** 3

**Excitement:**

4: Strong: This paper deepens the understanding of some phenomenon or lowers the barriers to an existing research direction.

**Missing References:**

L124: to my knowledge the MQM framework was introduced by Lommel et al. in 2014 (see https://doi.org/10.5565/rev/tradumatica.77)

**Paper Topic And Main Contributions:**

This work introduces InstructScore, a metric for text generation evaluation. Given a generated hypothesis and a reference text it generates an a list of error reports containing error type, location, severity, and an explanation. A score is assigned based on the severity of all errors.
This procedure is inspired by MQM annotations for evaluation. InstructScore is based on a LLaMa backbone which is fine-tuned on synthetic data generated by GPT-4. GPT-4 produces both the initial training data of hypotheses, references and error reports, as well as corrective feedback making sure that error reports are consistent. InstructScore is evaluated on a diverse set of text generation tasks and shows impressive performance.

**Questions For The Authors:**

A: from the formulation of the Exp-Generator (L157) is looks like the system will always produce a report consisting of at least 1 error even for error-free hypotheses. Is it possible for the system to determine that a hypothesis is error-free?

**Reasons To Accept:**

I think the idea of trying to produce MQM-style reports using an automated metric is really nice as it provides a lot more fine-grained information than just an opaque score.

The work shows a clever way to leverage the implicit knowledge of large language models for evaluation.

I think it is really impressive that the method consistently works well for a diverse range of text generation tasks (Table 2).

I appreciate the human study showing the alignment improvements in Fig. 7.

**Reasons To Reject:**

The procedure as described relies extremely heavily on synthetic data from GPT4. I think a reasonable question is whether one couldn't just prompt GPT4 to produce error reports directly, which I think might have been a good baseline for comparison.

The exact procedure for collecting synthetic data is very hard to understand without carefully studying the Appendix.

Overall the described process is very involved and I am happy that the code is already available, as the process of reproducing this seems very labor-intensive.

I think the explanation aspect of the work is a little under-explored. The error reports include explanations but it is not clear how helpful they will be in practice. Indeed, the human evaluation shows that they are wrong in many cases (Fig. 7).

**Reproducibility:**

2: Would be hard pressed to reproduce the results. The contribution depends on data that are simply not available outside the author's institution or consortium; not enough details are provided.

**Reviewer Confidence:**

3: Pretty sure, but there's a chance I missed something. Although I have a good feel for this area in general, I did not carefully check the paper's details, e.g., the math, experimental design, or novelty.

---

> ### Author Rebuttal · Authors · 2023-08-28
>
> Thank you very much for your feedback on our work!
>
> Q.A: Error-free cases?
>
> Answer: To address the question of the "no-error" case, our InstructScore can produce a no-error result if there are no major and minor errors present.
>
> Q.B: Direct error report using GPT-4?
>
> Answer: In our preliminary experiments, we found that just prompting GPT-4 is far from optimal and would clearly underperform our model. Similar behavior has also been reported by (https://arxiv.org/pdf/2303.13809.pdf). Moreover, the experiments of running GPT-4 over our entire testing set at WMT22 will cost around 1500 US dollars and 5000 US dollars for all five testing sets, which exceeds our research budget. Instead we include the existing best performed GPT-4 results (GEMBA-GPT4) on WMT22’s testing set to demonstrate our performance against prompt-based GPT4 methods. We would like to clarify that our approach does not heavily rely on GPT-4. Instead the goal is to finetune a small LLM to overcome the limitation of GPT-4.
>
> Q.C: Organization of the pipeline
>
> We will improve the description of the data construction pipeline and move some parts of the appendix to the main body of the paper.
>
> Q.D: Explanation aspect of InstructScore?
>
> We conduct human evaluation on the quality of explanations (with three annotators). We calculated precision and recall of the explanations before and after refinement. Precision = the number of correctly annotated error fields / total number of instructScore's labeled error fields. Recall = the number of correctly annotated error fields / (the number of correctly annotated error fields+the number of error fields that InstructScore missed). Based on the human evaluation result, 89.4% of InstructScore’s output after refinement is correct and it can identify 85.6% of errors produced by the translation system. To the field of explanation in particular, 86.1% of InstructScore’s output after refinement is correct according to humans and it can explain 85.0% of errors produced by the translation system. We will add these numbers to the revision.
>
> | All fields | Before refinement | After refinement |
> | ---------- | ----------------- | ---------------- |
> | Precision  | 0.778             | 0.894            |
> | Recall     | 0.824             | 0.856            |
>
> | Explanation field | Before refinement | After refinement |
> | ----------------- | ----------------- | ---------------- |
> | Precision         | 0.756             | 0.861            |
> | Recall            | 0.819             | 0.850             |

---

### Meta-Review · Area_Chair_11Et · 2023-09-23

**Recommendation:** 4

**Metareview:**

This paper introduces a metric that attempts to identify errors in generated text using LLaMA-7B. The authors show that their metric works across a variety of tasks and also validate with human judgments.

Reviewers uTpq and p5F9 were both impressed with the idea of using GPT-4 to help fine-tune LLaMA. Reviewer uTpq also points out that there's very nice work to show the alignment with human judgments. In the rebuttal the authors allayed some of the issues that the reviewers had by explicitly stating that they will clarify the pipeline and include the refinement table.

---

### Decision · Program_Chairs · 2023-10-07

**Decision:**

Accept-Main

**Comment:**

This paper introduces a metric that attempts to identify errors in generated text using LLaMA-7B. The authors show that their metric works across a variety of tasks and also validate with human judgments.

Reviewers uTpq and p5F9 were both impressed with the idea of using GPT-4 to help fine-tune LLaMA. Reviewer uTpq also points out that there's very nice work to show the alignment with human judgments. In the rebuttal the authors allayed some of the issues that the reviewers had by explicitly stating that they will clarify the pipeline and include the refinement table.